# It Has Potential: Gradient-Driven Denoisers for Convergent Solutions to Inverse Problems

**Regev Cohen**
Verily Research, Israel
regevcohen@google.com

**Yochai Blau**
Google Research, Israel
yochaib@google.com

**Daniel Freedman**
Verily Research, Israel
danielfreedman@google.com

**Ehud Rivlin**
Verily Research, Israel
ehud@google.com

## Abstract

In recent years there has been increasing interest in leveraging denoisers for solving general inverse problems. Two leading frameworks are regularization-by-denoising (RED) and plug-and-play priors (PnP) which incorporate explicit likelihood functions with priors induced by denoising algorithms. RED and PnP have shown state-of-the-art performance in diverse imaging tasks when powerful denoisers are used, such as convolutional neural networks (CNNs). However, the study of their convergence remains an active line of research. Recent works derive the convergence of RED and PnP methods by treating CNN denoisers as approximations for maximum a posteriori (MAP) or minimum mean square error (MMSE) estimators. Yet, state-of-the-art denoisers cannot be interpreted as either MAP or MMSE estimators, since they typically do not exhibit symmetric Jacobians. Furthermore, obtaining stable inverse algorithms often requires controlling the Lipschitz constant of CNN denoisers during training. Precisely enforcing this constraint is impractical, hence, convergence cannot be completely guaranteed. In this work, we introduce image denoisers derived as the gradients of smooth scalar-valued deep neural networks, acting as potentials. This ensures two things: (1) the proposed denoisers display symmetric Jacobians, allowing for MAP and MMSE estimators interpretation; (2) the denoisers may be integrated into RED and PnP schemes with backtracking step size, removing the need for enforcing their Lipschitz constant. To show the latter, we develop a simple inversion method that utilizes the proposed denoisers. We theoretically establish its convergence to stationary points of an underlying objective function consisting of the learned potentials. We numerically validate our method through various imaging experiments, showing improved results compared to standard RED and PnP methods, and with additional provable stability.

## 1 Introduction

**General Recovery** Inverse problems aim to recover an unknown signal from its observed corrupted or compressed version. This fundamental task appears in a variety of fields such as machine learning, signal processing, computer vision and medical imaging. When a model for the degradation process is available, one may derive a data fidelity term which relates the measurements to the unknown signal through a forward operator. However, this term alone typically leads to ill-posed optimization problems with unstable solutions, requiring the incorporation of additional knowledge of the desired

35th Conference on Neural Information Processing Systems (NeurIPS 2021).

signal into the inversion process. This has given rise to diverse regularization methods which have been investigated extensively over the last few decades.

**RED and PnP**  A very basic inverse problem is denoising, in which a clean signal is to be recovered from its measurements contaminated with additive noise. During the last period of time there has been an overwhelming advancement in the ability to perform denoising, creating techniques that approach optimal performance [11, 29, 30]. This striking progress in denoising has been the motivation for the plug-and-play priors (PnP) [48, 10, 41] methodology, which introduced the paradigm of using denoisers in order to solve general inverse problems. PnP incorporates implicit regularization into the inverse solutions by replacing proximal operators in common optimization algorithms with powerful denoisers. This simple modification has led to remarkable performance in diverse inverse tasks [41, 50, 38, 22, 56, 3, 51], as well as to the development of an arsenal of PnP-based techniques which includes proximal-type methods but also goes beyond them [33, 7, 18, 15, 5, 55, 14, 17]. However, the lack of a clearly formulated regularization term corresponding to the denoiser poses a serious challenge in ensuring the stability of PnP-based methods, and typically requires careful parameter-tuning [49].

To fill this void, the regularization by denoising (RED) [39] framework presents an explicit Laplacian-based regularization that relies on a denoiser. Under certain conditions on the denoiser [39, 37], the RED functional is convex and its gradient is given by the denoising residual, allowing for easy incorporation in standard optimization procedures. RED has demonstrated impressive recovery performance while providing global convergence guarantees. However, various common denoisers do not satisfy the conditions required by RED [37]. In such cases, there exists no objective function which justifies the RED method [37] and the corresponding convergence analysis does not hold. Thus RED, as well as PnP, shows state-of-the-art empirical results but without provable stability. A comprehensive overview of PnP and RED is given in [3].

**Related Work**  The study of the stability of both RED and PnP methods has become an active line of research in recent years. The convergence of two PnP techniques was proven in [10] and [47] under the assumption of bounded denoisers. When minimum mean squared error (MMSE) denoisers are used, the PnP framework was shown to converge vis-a-vis an explicit objective function [53]. Similarly, [7, 18] proved the convergence of approximate message passing algorithms with MMSE denoisers for random froward operators. Multiple studies [41, 43, 44, 9, 45] derived the convergence of PnP by relying on nonexpansive denoisers. The latter has been the core assumption of [8], which explained the PnP framework, including its stability, through the notion of consensus equilibrium. A similar condition was used in [37], which adopted the score-matching framework to provide a new interpretation of the RED algorithms and their convergence. [42] presented a block-coordinate RED method and derived its fixed-point convergence for block-nonexpansive denoisers. The work of [36] introduced a convergent PnP-type method using a maximally monotone denoiser, created by training a nonexpansive neural network in a supervised fashion. Alternatively, [40] provided convergence guarantees for certain PnP variants that use trained denoisers whose residuals are nonexpansive. This type of denoiser was used in [27] to develop Bayesian inference with PnP priors, including the extension of the PnP framework to Monte Carlo sampling. The authors of [24, 23] employ denoisers within Langevin dynamics to sample from the posterior distribution of any linear inverse problem. A different strategy was studied in [35], which proposed a special inner product derived from the given linear denoiser, and thus proved the convergence of PnP based on the theory of averaged nonexpansive operators. Recent work [12] introduced RED via fixed-projection framework (RED-PRO), which utilizes the fixed-points of the denoisers as regularization. RED-PRO unifies RED and PnP and derives their convergence for general demicontractive denoisers, including nonexpansive ones.

Recent studies, mentioned above, are motivated by or rely on the assumption that the denoiser is an MMSE or maximum a posteriori (MAP) estimator. However, such estimators are in the form of gradients, hence, they must exhibit symmetric Jacobians [20, 19, 21] amongst other properties. Furthermore, Jacobian symmetry offers additional advantages as it often leads to stable algorithms and allows one to better understand the behavior of such non-linear estimators [34]. Unfortunately, many state-of-the-art denoisers do not fulfill this condition [37]. An additional drawback of current RED and PnP methods is that the Lipschitz constant of the denoiser often has to be controlled to create a nonexpansive operator, which in turn ensures convergence. This is typically done by learning a deep network and adding a regularization term to the training loss function, whose purpose is to bound the Lipschitz constant of the resultant denoiser. However, exactly enforcing this property of the denoiser is impractical and it is only approximated in practice.

**Contributions** In this work, we develop a provably convergent technique which exploits denoisers for solving general inverse problems. (1) We introduce image denoisers that are constructed as the gradients of smooth potential functions, represented by deep neural networks and trained in a supervised fashion. (2) Equipped with these learned potential networks, we formulate a general inverse framework where we utilize our denoising potentials as an explicit regularization. (3) We use the existence of an underlying objective function to incorporate the proposed denoisers into standard optimization schemes with backtracking step size. Specifically, we present a simple gradient descent algorithm where the step size is tuned automatically to provably guarantee convergence to stationary points of the objective function. (4) We numerically validate our method through imaging experiments, showing improved results compared to RED and PnP, with additional provable stability.

## 2 Background

**Inverse Problems: Formulation** We consider the recovery of an unknown signal $\mathbf{x}^* \in \mathbb{R}^{d_x}$ from its corrupted observation $\mathbf{y} \in \mathbb{R}^{d_y}$. The inversion task is traditionally formulated as an optimization problem of the form

$$\hat{\mathbf{x}} = \underset{\mathbf{x} \in \mathbb{R}^{d_x}}{\arg\min} \ f(\mathbf{x}; \mathbf{y}) + \lambda g(\mathbf{x}), \tag{1}$$

where $f(\cdot\,; \mathbf{y})$ is a fidelity term related to the observations $\mathbf{y}$, $g(\cdot)$ is a regularization functional that incorporates prior information or known statistics of the true signal $\mathbf{x}^*$ to avoid ill-posed problems, and $\lambda > 0$ balances between the terms. A common model for the measurements is $\mathbf{y} = \mathbf{A}\mathbf{x} + \mathbf{w}$ where $\mathbf{A}$ is a linear degradation (forward) operator and $\mathbf{w}$ is an i.i.d. white Gaussian noise, leading to the following log-likelihood function

$$f(\mathbf{x}; \mathbf{y}) = \frac{1}{2}||\mathbf{A}\mathbf{x} - \mathbf{y}||_2^2. \tag{2}$$

The above model applies to a wide variety of inverse problems such as compressed sensing, super resolution, deblurring, inpainting and much more.

In the following, we consider a general functional $f(\cdot)$ where we suppress its dependency on the observations $\mathbf{y}$ for brevity. Moreover, both $g(\cdot)$ and $f(\cdot)$ are typically assumed to be convex where the former is generally non-differentiable, while the latter is a smooth function. By the definition of smooth functions, there exists a Lipschitz constant $L_f > 0$ such that the gradient $\nabla f$ of $f(\cdot)$ satisfies

$$||\nabla f(\mathbf{x}_1) - \nabla f(\mathbf{x}_2)|| \le L_f ||\mathbf{x}_1 - \mathbf{x}_2||, \ \forall\, \mathbf{x}_1, \mathbf{x}_2 \in \mathbb{R}^{d_x}. \tag{3}$$

**Proximal Methods** The minimization problem (1) has been investigated extensively over the decades, leading to a series of diverse approaches. Many common optimization techniques for solving (1) rely on the proximal operator of $g(\cdot)$, defined as

$$\mathcal{P}_{\lambda g}(\mathbf{x}) \triangleq \underset{\mathbf{z} \in \mathbb{R}^{d_x}}{\arg\min} \ \frac{1}{2}||\mathbf{x} - \mathbf{z}|| + \lambda g(\mathbf{z}). \tag{4}$$

The proximal operator, which is a MAP estimator, provides an efficient way to overcome the lack of differentiablity of $g(\cdot)$, especially when it admits a closed-form expression, for example in the case of the $\ell_1$-norm. A simple representative of this family of algorithms is the proximal gradient descent (PGD) method which updates the solution iteratively as follows

$$\mathbf{x}_{k+1} = \mathcal{P}_{\lambda g}\Big(\mathbf{x}_k - \mu_k \nabla f(\mathbf{x}_k)\Big) \tag{5}$$

where $\mu_k > 0$ is the step size. PGD has shown great performance in various applications along with strong convergence guarantees.

**The PnP Framework** Notice that the proximal operator can be seen, by definition, as the solution to a denoising problem. This observation serves as the basis of the PnP framework which extends standard proximal methods by replacing the proximal operator with powerful denoisers. Following this rationale, (5) is modified to create the PnP-PGD method that performs the following update rule

$$\mathbf{x}_{k+1} = \mathcal{D}\Big(\mathbf{x}_k - \mu_k \nabla f(\mathbf{x}_k)\Big), \tag{PnP-PGD}$$

where $\mathcal{D} : \mathbb{R}^{d_x} \to \mathbb{R}^{d_x}$ is an off-the-shelf denoiser.

**The RED Framework** An alternative strategy for leveraging advanced denoisers is the RED framework which utilizes the denoiser to construct an explicit regularization $g(\mathbf{x}) = \frac{1}{2}\langle \mathbf{x}, \mathbf{x} - \mathcal{D}(\mathbf{x})\rangle$. Under certain conditions on the denoiser [39, 37], $g(\cdot)$ is convex and its gradient is readily given by the denoising residual $\nabla g(\mathbf{x}) = \mathbf{x} - \mathcal{D}(\mathbf{x})$. This allows for the solution of (1) by standard algorithms, e.g., steepest (or gradient) descent

$$\mathbf{x}_{k+1} = \mathbf{x}_k - \frac{\mu_k}{1+\lambda}\Big(\nabla f(\mathbf{x}_k) + \lambda\big(\mathbf{x}_k - \mathcal{D}(\mathbf{x}_k)\big)\Big). \tag{RED-SD}$$

Surprisingly, RED-based methods have been found to be effective also when the required conditions on the denoiser are not satisfied and the expression for the regularization term is invalid. In such cases, RED uses the denoiser as an implicit regularization, similar to PnP.

**Challenges in RED and PnP** Both RED and PnP provide extremely efficient means for exploiting powerful denoisers to solve inverse problems. Yet their analysis poses two closely related challenges:

1. *Asymmetric Denoisers:* Many studies on RED and PnP [53, 27, 43, 44, 9, 45, 52, 42] motivate or base their derivations on the assumption that the denoiser is an approximation of either MAP or MMSE estimator. It has been proven that when the Jacobian of such estimators is well-defined it must be symmetric [20, 19, 21]. However, RED and PnP typically reach state-of-the-art results by relying on denoisers which do not exhibit this property, as in the case of the popular DnCNN technique; as a result, they lose their interpretation as optimization solvers. In such cases, while there are alternative optimization-free explanations for RED and PnP [8, 37], it is challenging to properly analyze their behavior and characterize the nature of their solutions.

2. *Convergence Issues:* Continuing the above line of reasoning, often there exists no regularization that explains the denoiser, namely no functional $g(\cdot)$ for which $\nabla g(\mathbf{x}) = \mathcal{D}(\mathbf{x})$ or $\nabla g(\mathbf{x}) = \mathbf{x} - \mathcal{D}(\mathbf{x})$ [37]. This lack of a potential function creates difficulties in studying the theoretical convergence of the RED and PnP algorithms. However, a common sufficient condition to guarantee convergence is for the denoiser $\mathcal{D}(\cdot)$ to be (averaged) nonexpansive [13], implying that

$$||\mathcal{D}(\mathbf{x}_1) - \mathcal{D}(\mathbf{x}_2)|| \leq L||\mathbf{x}_1 - \mathbf{x}_2||, \ \forall \mathbf{x}_1, \mathbf{x}_2 \in \mathbb{R}^{d_x}, \tag{6}$$

for some Lipschitz constant $0 < L \leq 1$. However, most predefined denoisers do not satisfy this assumption. An alternative approach [40, 36, 46] is to learn a nonexpansive denoiser by adding a regularization term to the training loss which constrains its Lipschitz constant. Unfortunately, this remains an open challenge since in general there are no practical means to absolutely enforce the resultant denoiser to be nonexpansive.

Our work addresses both of these challenges. We first introduce a new class of denoisers which stem from an explicit potential functional and thus demonstrate symmetric Jacobian by their construction. We then derive an inverse technique that utilizes our denoisers as well as their potentials to ensure convergence. Three important remarks are to be made with regard to work presented below. First, we emphasize that the proposed denoisers are not intended to compete with state-of-the-art denoisers, but rather are developed to exhibit favorable properties that benefit other inversion tasks. Second, our proposed recovery method is not derived to compete with end-to-end learned solutions to specific inverse problems [31, 2, 25, 26]. Rather, our learned denoisers, along with our optimization framework, can be used for *all* inverse problems (of the form in Eq. (1)) without additional training. Lastly, while we present a present a specific inversion method, the potential-driven denoisers can benefit many RED and PnP algorithms, hence, our work complements these frameworks.

## 3 Recovery by Potential-Driven Denoisers

In this section, we present our denoisers driven by learned potential functions. Designed as gradient operators, the denoisers display symmetric Jacobians, allowing for a MAP or an MMSE estimator interpretation. More importantly, we use the denoising potentials as an explicit regularization for general inverse problems, leading to provably convergent optimization schemes which utilize our proposed denoisers. Unlike previous works, we do not require bounding or approximation of the Lipschitz constants of the denoisers.

### 3.1 Potential-Driven Denoisers

**MAP and MMSE Denoisers as Gradients** Two fundamental forms of denoisers are MAP and MMSE estimators. The theoretical properties of MAP and MMSE denoisers are well-understood and

they serve as the basis of many works which study the convergence of RED and PnP methods. The following lemmas reveals that MAP and MMSE denoisers share a common structure:

**Lemma 1** (Corollary 1 [21]). *Assume the denoiser is a proximal operator $\mathcal{D} \triangleq \mathcal{P}_\phi$ of some scalar-valued function $\phi$. Then, there exists a convex scalar-valued function $\psi(\cdot)$ such that $\mathcal{D} = \nabla\psi$.*

**Lemma 2** (Tweedie's Formula [16]). *Assume the denoiser is an MMSE estimator $\mathcal{D}$ of images contaminated with white Gaussian noise of variance $\sigma^2$. Then, $\mathcal{D} \triangleq I + \sigma^2\nabla\log p(\cdot)$ where $I$ is the identity operator and $p(\cdot)$ denotes a smooth probability density of images.*

The above results imply that for any MAP or MMSE denoiser $D$ there exists a scalar-valued function $\rho$ such that the denoiser, or equivalently its residual, can be expressed as $D = \nabla\rho$. This observation sets the main foundation of our gradient-driven denoisers, described next.

**Construction of Gradient-Based Denoisers**   Now, we consider the task of training a deep network to perform denoising. While we focus on image denoising, the following derivations can be generalized to any signal of interest. Motivated by the above observation, we design a denoising deep network which is the gradient of scalar-valued function, acting as a potential function. Three possible approaches to construct gradient-driven denoisers are as follows:

1. Directly design a deep network which is the gradient of a certain potential function. This approach avoids building explicitly the potential network. In the Supplemental we describe certain conditions under which a given network is a gradient.

2. Compose a potential function using an arbitrary deep network $N_\theta$ whose input is an image. Simple examples of this approach are $\langle \mathbf{x}, \ N_\theta(\mathbf{x})\rangle$, $||N_\theta(\mathbf{x})||^2$, $||\mathbf{x} - N_\theta(\mathbf{x})||^2$, etc. Note that in general the network $N_\theta$ does not equal to the gradient of the composite potential.

3. Model the potential function as scalar-valued deep network and use its gradient as the denoiser.

Here we focus on the last approach since it is the most general. We construct a potential function $g_\theta(\cdot) \in \mathbb{R}$ defined as a deep neural network with parameters $\theta$. As illustrated in Fig. 1, the potential network follows a typical structure of feed-forward network with skip-connections except from two main modifications:

- The output of the network is a single scalar.
- The acitvations are continuously differentiable and smooth functions.

Mathematically, the potential network consists of $M > 0$ layers described recursively as follows

$$\mathbf{z}_{m+1} = s_m\big(\mathcal{W}_m\mathbf{z}_m + \mathcal{V}_m\mathbf{x} + b_m\big), \ m = 0, ..., M - 1. \tag{7}$$

Here, $\mathbf{x} \in \mathbb{R}^{d_x}$ is the input noisy image, $\mathbf{z}_m \in \mathbb{R}^{d_m}$ are the hidden variables where $\mathbf{z}_0 \equiv 0$ and $g_\theta(\mathbf{x}) = \mathbf{z}_M$ is the output of the network. The parameters of the network are $\theta \triangleq \{b_m, \mathcal{W}_m, \mathcal{V}_m, \}_{m=0}^{M-1}$ where $b_m$ is a bias term and $\mathcal{W}_m, \mathcal{V}_m$ are linear operators with $\mathcal{W}_0 \equiv 0$. The activation functions are denoted by $s_m : \mathbb{R} \to \mathbb{R}$ and applied element-wise.

To allow the network to accept inputs of variable sizes, we choose the linear mappings to be standard convolutions. We ensure the output of the network is a scalar by setting the number of output channels of the last convolution to be one, and using global average pooling as the final activation function. Another important design choice is that all activations are continuously differentiable and smooth functions. This creates a potential network that is smooth as the composition of smooth functions, which is crucial for guaranteeing convergence of our inversion method, presented in Section 3.2.

Given the potential function $g_\theta(\cdot)$, we define our denoiser $\mathcal{D}_\theta(\cdot)$ as

$$\mathcal{D}_\theta(\mathbf{x}) \triangleq \mathbf{x} - \nabla g_\theta(\mathbf{x}). \tag{8}$$

Alternatively, we can write or $\mathcal{R}_\theta = \nabla g_\theta$ where $\mathcal{R}_\theta$ is the denoising residual. This construction ensures that the resultant denoiser, termed gradient-driven denoising CNN (GraDnCNN), is indeed a gradient of an explicit function, and thus it possesses a symmetric Jacobian. This form of gradient-driven denoisers represents more faithfully MAP and MMSE estimators, hopefully leading to better

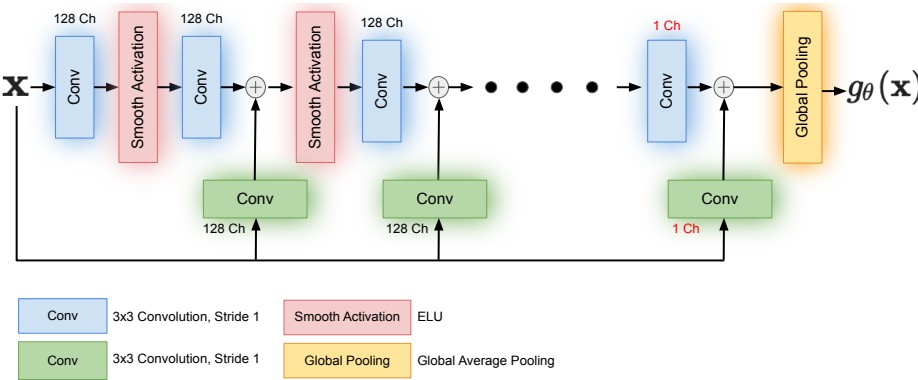

Figure 1: Network Architecture.

approximations. Finally, we perform supervised residual learning and train the denoiser (or equivalently the potential) by minimizing a chosen loss function $\mathcal{L}(\cdot, \cdot)$ with respect to the parameters

$$\theta^* = \arg\min_{\theta} \sum_n \mathcal{L}\Big(\mathbf{x}_n - \mathbf{x}_n^*, \mathcal{R}_\theta(\mathbf{x}_n)\Big),\tag{9}$$

where $\{\mathbf{x}_n^*, \mathbf{x}_n\}$ are pairs of clean and noisy images respectively.

**Special Potentials: Convex Functions and Difference of Convex Functions**  An additional interesting path we follow is setting all $\{\mathcal{W}_m\}$ (blue blocks in Fig. 1) to be non-negative and all activation functions $\{s_m\}$ (red blocks in Fig. 1) to be convex and non-decreasing mappings. This creates a potential function $g_\theta(\cdot)$ which is an input convex neural network (ICNN) [4], leading to a denoiser called denoising ICNN (DnICNN). As we explain later in Section 3.2, the use of convex potentials leads to convex inverse problems, thus it facilitates the optimization process and allows to derive recovery methods with global convergence guarantees.

The down side of convex potentials is that the non-negative weights $\{\mathcal{W}_m\}$ may limit the expressivity of the network. Hence, we propose an alternative where the potential network is defined as the difference of two ICNNs

$$g_\theta(\mathbf{x}) \triangleq g_{\theta_1}(\mathbf{x}) - g_{\theta_2}(\mathbf{x}).\tag{10}$$

The resultant potential enjoys favorable properties as the difference of two convex functions [28] and it can approximate non-convex functions (unlike the convex potentials). Thus, we present a third denoiser, referred to as denoising by difference of ICNNs (DnDICNN), which acts as a compromise between GraDnCNN and DnICNN. In the following section, we study the properties of these three denoisers in the context of our recovery scheme, both in terms of stability and recovery performance.

## 3.2    Convergent Inversion Method

**Inversion by Descent**  Given a trained gradient-driven denoiser, we wish to exploit the denoising power within general recovery processes. The existence of an underlying learned potential $g_\theta(\cdot)$ allows us to formulate the recovery as the following problem:

$$\hat{\mathbf{x}} = \arg\min_{\mathbf{x} \in \mathbb{R}^{d_x}} f(\mathbf{x}) + \lambda g_\theta(\mathbf{x}).\tag{11}$$

Here we assume $f(\cdot)$ is a continuously differentiable smooth function, typically the log-likelihood of the observations given by (2). The potential network acts as a regularization term or a smooth image prior which creates a well-posed problem. Furthermore, by our construction the potential $g_\theta(\cdot)$ is smooth, thus, the complete objective, denoted by $F(\cdot)$, is smooth with some unknown Lipschitz constant $L > 0$. This facilities the optimization process considerably and we can solve (11) via a simple gradient descent algorithm

$$\mathbf{x}_{k+1} = \mathbf{x}_k - \mu_k \nabla F(\mathbf{x}_k),\tag{12}$$

where we start with some initial solution $\mathbf{x}_0$. Notice the above iteration can be written as

$$\mathbf{x}_{k+1} = \mathbf{x}_k - \mu_k \Big(\nabla f(\mathbf{x}_k) + \lambda\big(\mathbf{x}_k - \mathcal{D}_\theta(\mathbf{x}_k)\big)\Big)\tag{13}$$

where $\mathcal{D}_\theta(\cdot)$ is our denoiser originating from $g_\theta(\cdot)$.

**Backtracking Step Size**   Note that the step size $\mu_k$ must be set properly to reach convergence. Typically, one only has access to the denoiser itself, hence the step size is tuned empirically which may lead either to divergence or to slow convergence due to an overly small step size. By contrast, we have complete access to the objective function, thus, we can properly set the step size to ensure convergence. To that end we rely on the Descent Lemma [6] in the Supplemental, which implies that

$$F(\mathbf{x}_{k+1}) \leq F(\mathbf{x}_k) - \frac{\mu_k}{2}||\nabla F(\mathbf{x}_k)||_2^2, \ \forall \mu_k : 0 \leq \mu_k \leq \frac{1}{L}. \tag{14}$$

When $\nabla F(\mathbf{x}_k) \neq 0$ we obtain that $F(\mathbf{x}_{k+1}) < F(\mathbf{x}_k)$; otherwise, we reach a critical point. The challenge that remains is setting the step size according the Lipschitz constant $L$ which is typically unknown. To address this, we set a backtracking step size defined as $\mu_k \triangleq 1/\hat{L}$ where $\hat{L}$ is an approximation of the true constant $L$. We start with some initial guess $\hat{L} = L_0$ and at each iteration update $\hat{L}$ and $\mu_k$ accordingly until inequality (14) is satisfied, ensuring we perform a descent step.

---

**Algorithm 1:** Regularization by Potential-Driven Denoising

---

**Input:** initial point $\mathbf{x}_0 \in \mathbb{R}^{d_x}$, $L_0 > 0$, $\beta > 1$, $\lambda > 0$, $f(\cdot)$, $g_\theta(\cdot)$ and the related denoiser $\mathcal{D}_\theta(\cdot)$.

1 Set $F(\cdot) \triangleq f(\cdot) + \lambda g_\theta(\cdot)$ and $\hat{L} = L_0$.
2 **for** *k=0,1,2,...* **do**
3 $\quad$ $\mu_k = 1/\hat{L}$.
4 $\quad$ $\mathbf{x}_{k+1} = \mathbf{x}_k - \mu_k \Big( \nabla f(\mathbf{x}_k) + \lambda \big( \mathbf{x}_k - \mathcal{D}_\theta(\mathbf{x}_k) \big) \Big)$.
5 $\quad$ **while** $F(\boldsymbol{x}_{k+1}) > F(\boldsymbol{x}_k) - \frac{\mu_k}{2}||\nabla F(\boldsymbol{x}_k)||_2^2$ **do**
6 $\quad\quad$ $\hat{L} = \beta \cdot \hat{L}, \mu_k = 1/\hat{L}$.
7 $\quad\quad$ $\mathbf{x}_{k+1} = \mathbf{x}_k - \mu_k \Big( \nabla f(\mathbf{x}_k) + \lambda \big( \mathbf{x}_k - \mathcal{D}_\theta(\mathbf{x}_k) \big) \Big)$.

---

The resulting inversion technique, outlined in Algorithm 1, can be viewed as RED-SD with backtracking step size which is enabled by the existence of an available prior function $g_\theta(\cdot)$. In addition, note that throughout the iterations it holds that $\hat{L} \leq \max(L_0, \beta L)$ where $L$ is the true unknown Lipschitz constant. Also note that the total number of updates of $\hat{L}$ is bounded by $\log_\beta(L/L_0) \approx \mathcal{O}(\log_2 L)$.

**Convergence Guarantees**   We remark that other more sophisticated optimization techniques can be considered for solving (11), including acceleration schemes. However, rather than providing the most efficient recovery method, we focus here on simplicity to highlight the benefits of learned potential-driven denoisers. Moreover, there is a vast literature which provides convergence guarantees for Algorithm 1 under various conditions. In the following theorem we present a standard convergence analysis based on the smoothness of the objective function. The proof is given in the Supplemental.

**Theorem 1.** *Assume that $f(\cdot)$ and $g_\theta(\cdot)$ are continuously differentiable and smooth functions. Define $F(\cdot) \triangleq f(\cdot) + \lambda g_\theta(\cdot)$ and assume it is bounded below by $F^*$. Let $\{\boldsymbol{x}_k\}_{k \geq 0}$ be the sequence generated by Algorithm 1. Then,*

$$\min_{m=0,1,...,k-1} ||\nabla F(\boldsymbol{x}_m)||_2^2 \leq \frac{2\max(L_0, \beta L)}{k}\big(F(\boldsymbol{x}_0) - F^*\big).$$

The above theorem implies that $||\nabla F(\mathbf{x}_k)||_2^2 \to 0$ as $k \to \infty$ and that all limit points of the sequence $\{\mathbf{x}_k\}_{k \geq 0}$ are stationary points of the objective function $F(\cdot)$. Notice this result applies to all the three denoisers (or potentials) presented in Section 3.1. This emphasizes the importance of the smoothness of the potential functions which ensures the convergence of the recovery, regardless to the highly non-convex nature of the objective function.

Commonly, the function $f(\cdot)$ is a convex function, e.g. when it is the log-likelihood. In such scenarios, the use of convex potential networks leads to a special case where the recovery problem is convex and we obtain global convergence guarantees:

**Corollary 1.** *Assume that $f(\cdot)$ and $g_\theta(\cdot)$ are convex in addition to the conditions of Theorem 1. Then, Algorithm 1 converges to global minima of the objective function $F(\cdot)$.*

The proof is straightforward since according to Theorem 1 the sequence generated by Algorithm 1 converges to stationary points of the objective function, and any stationary point of a convex function

is a global minimizer. Thus, in the context of inverse problems, our convex potential networks connect the well-known world of convex optimization and with the highly non-linear non-convex nature of deep neural networks.

Finally, for the completeness of our work, we study the case where the recovery problem includes additional constraints on the solution, leading to the following formulation

$$\hat{\mathbf{x}} = \arg\min_{\mathbf{x} \in \mathbb{R}^{d_x}} f(\mathbf{x}) + \lambda g_\theta(\mathbf{x}) + h(\mathbf{x}), \tag{15}$$

where the function $h(\cdot)$ is convex yet possibly non-smooth. We provide the full analysis consisting of algorithmic and theoretical results in the Supplemental.

# 4 Numerical Experiments

Here we study the performance of Algorithm 1 using our three proposed potential-driven denoisers: GraDnCNN, DnICNN, DnDICNN. We compare ourselves to PnP-PGD and RED-SD, applied with the popular DnCNN denoiser [54], for the tasks of Gaussian deblurring and image super resolution.

**Datasets**   For training the denoising networks for blind Gaussian denoising we use the public DIV2K dataset [2] that consists of a total of 900 high resolution images, 800 for training and 100 for validation. We perform data augmentation by randomly cropping 100 patches of size $50 \times 50$ from each image of the DIV2K training set. We then contaminate each patch with an additive i.i.d. white Gaussian noise (WGN) with a noise level $\sigma_n$ chosen uniformly at random from the range $[1, 55]$, creating our 80,000 sample training set. For our validation set, we replicate each image of the DIV2K validation set 10 times, and we add WGN to each copy with random noise level in the specified range. As our test images for evaluating the inverse methods for deblurring and super resolution, we use two different datasets. The first set, provided by the authors of [39], consists of 11 common color images (butterfly, starfish, etc), and we refer to it as Set11. For a more through assessment, we utilize all 500 natural images from the public Berkeley segmentation dataset (BSD500) [32]. All pixel values of the images described above are in the range $[0, 255]$.

**Training Denoisers**   We experiment with four different denoisers. The first is a variant of the DnCNN-B architecture [54] with 7 layers, 128 feature maps, and $3 \times 3$ convolution kernels. For GraDnCNN we use the architecture shown in Fig. 1, consisting of $M = 6$ layers, ELU activation functions, 128 feature maps, and $3 \times 3$ bias-free convolutions where the final two convolution layers have a single output channel, and the last activation is global average pooling to produce a scalar. A similar architecture is used for DnICNN, where we set certain weights to be non-negative. The latter architecture is then used to construct our DnDICNN as the difference of two DnICNNs.

Given the datasets detailed above, we train each of the networks using an Adam optimizer for 100 epochs with a constant learning rate of $10^{-3}$. For the training loss, we use a modified version of mean squared error (MSE) cost function: $\sum_n \frac{1}{\sigma_n^2} \text{MSE}\left(\mathbf{x}_n - \mathbf{x}_n^*, \mathcal{R}_\theta(\mathbf{x}_n)\right)$ where $\mathbf{x}_n^*$ denotes a clean patch, $\mathbf{x}_n$ is the corresponding noisy patch contaminated with noise of level $\sigma_n$, and $\mathcal{R}_\theta(\mathbf{x}_n)$ is the output of the network. Since samples with larger noise levels typically lead to larger training errors, we find that this modification of dividing the MSE by the noise level is required to prevent the denoisers from over-smoothing. All experiments are performed in Tensorflow [1] where each model is trained on a single NVIDIA Tesla 32GB V100 GPU.

**Inverse Problems**   We focus here on two common inverse problems: Gaussian deblurring and single image super resolution. For the task of deblurring we perform the following process on the test datasets (Set11, BSD500) described above. Each image is convolved with a 2D Gaussian function $25 \times 25$ with a standard deviation of 1.6. Then, we add to the result an i.i.d. WGN of level $\hat{\sigma} = \sqrt{2}$. For super resolution we follow the same procedure except we add an intermediate step, before noise contamination, in which we scale-down the image by a factor of 3 at each axis.

**Recovery**   Given the degraded test images and the trained denoisers, we compare five different inverse methods: PnP-PGD and RED-SD, both with DnCNN, as well as Algorithm 1 employed with our three potential-driven denoisers. For Algorithm 1 we use the constrained update rule, described in the Supplemental, where the proximal operator is the projection of pixel values of the solution into the range $[0, 255]$. We apply the same projection on each intermediate solution of PnP-PGD

and RED-SD for improved stability. For properly setting the hyperparameters of the different tested methods we use Set11 to perform an extensive hyperparameter search, provided in the Supplemental.

We first study the convergence by running all the techniques for a fixed number of 400 iterations for both inverse tasks, where we measure convergence by $||\mathbf{x}_{k+1} - \mathbf{x}_k||/\mu_k$. The results, averaged over each dataset separately, are given in Figures 2 and 3. The foremost observation is that in all the scenarios PnP-PGD diverges at the a certain point. We conjecture the reason of this instability of PnP-PGD lies in the fact that unlike the other methods PnP-PGD includes no hyperparameters for constraining the denoiser, thus, powerful denoisers may lead to divergence by analogy with a too large learning rate. This can be prevented by considering more sophisticated version of PnP or by performing early stopping, however, it will introduce additional hyperparameter which need to be properly tuned for each data instance, making the technique less robust. Moreover, notice that the convergence of Algorithm 1 does not depend on the values of $\lambda$ and $\mu$, regardless of the potential-driven denoiser in use. This is a desired property that most RED and PnP methods do not exhibit. We note that the convex potential leads to fastest convergence which is expected, since in this case the objective function is smooth and convex, facilitating the optimization process significantly.

Next, we quantitatively compare the inverse algorithms by computing the average peak signal to noise ratio (PSNR) and structural similarity index (SSIM) of their solutions with respect to the ground truth images. As seen in Table 1, Algorithm 1 with GraDnCNN produces the best performance in both tasks and for both test sets. Algorithm 1 with DnICNN yields the worst results among our potential-driven denoisers, yet comparable to PnP-PGD and RED-SD with DnCNN. The performance of Algorithm 1 with DnDICNN is consistent with the fact that DnDICNN presents a compromise between GraDnCNN and DnICNN in terms of the expressivity of the network. To visually support our claims, we provide selected recovered images in Fig. 4, while additional results are given in the Supplemental. An important remark is that in contrast to PnP and RED methods, presented here and many others, the solutions of Algorithm 1 and its performance are completely independed of the choice of the step-size $\mu$. Hence, Algorithm 1 is more robust.

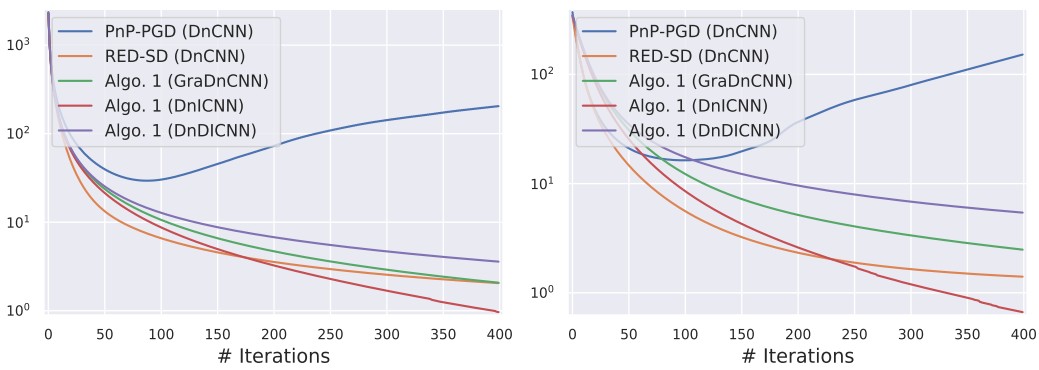

Figure 2: Average Convergence for Set11 - (Left) Gaussian deblurring (Right) Super resolution

Table 1: Average PSNR / SSIM obtained for different methods and datasets.

| Datasets | PnP-PGD DnCNN | RED-SD DnCNN | Algorithm 1 | | |
| --- | --- | --- | --- | --- | --- |
| | | | GraDnCNN | DnICNN | DnDICNN |
| Gaussian Deblurring | | | | | |
| **Set11** | 28.81 / 0.86 | 28.69 / 0.82 | 29.42 / 0.86 | 28.91 / 0.85 | 29.07 / 0.84 |
| **BSD500** | 28.79 / 0.84 | 28.32 / 0.79 | 28.91 / 0.83 | 28.63 / 0.82 | 28.41 / 0.80 |
| Super Resolution | | | | | |
| **Set11** | 26.20 / 0.77 | 26.14 / 0.76 | 26.69 / 0.79 | 25.95 / 0.76 | 26.20 / 0.76 |
| **BSD500** | 26.36 / 0.75 | 26.24 / 0.73 | 26.57 / 0.74 | 26.15 / 0.73 | 25.95 / 0.71 |

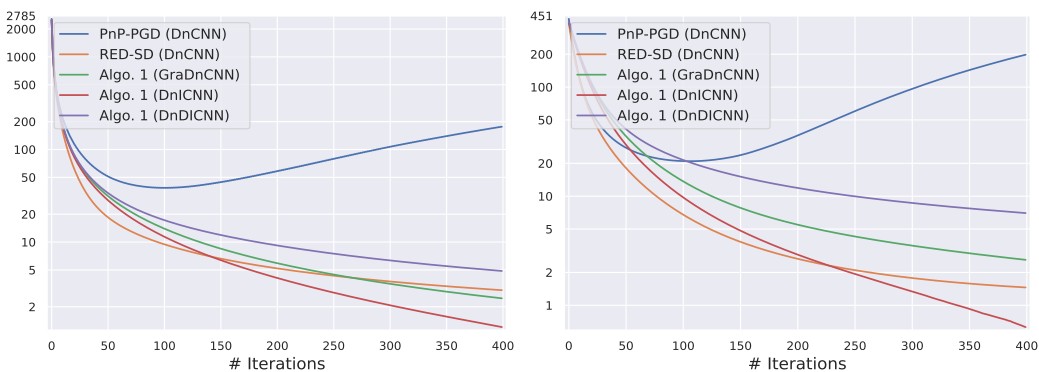

Figure 3: Average Convergence for BSD500 - (Left) Gaussian deblurring (Right) Super resolution.

# 5 Conclusion

In this paper we developed a provably convergent method with explicit denoising regularization for solving inverse problems. To that end, we first introduced the notion of potential-driven denoisers, i.e. denoisers which are based on the gradient of learned scalar-valued networks. We then utilized our denoisers along with their potential functions to derive an inverse gradient descent method with backtracking step size, ensuring convergence. We demonstrate experimentally that the resulting methods provide improved performance compared to standard RED and PnP methods, while offering provable stability.

The developed framework complements RED and PnP as our potential-driven denoisers can be used within any RED or PnP techniques to guarantee convergence, regardless of the data being used. Furthermore, while our work focus on learning denoisers, the concept of potential-driven networks may be generalized to benefit other image-to-image operations.

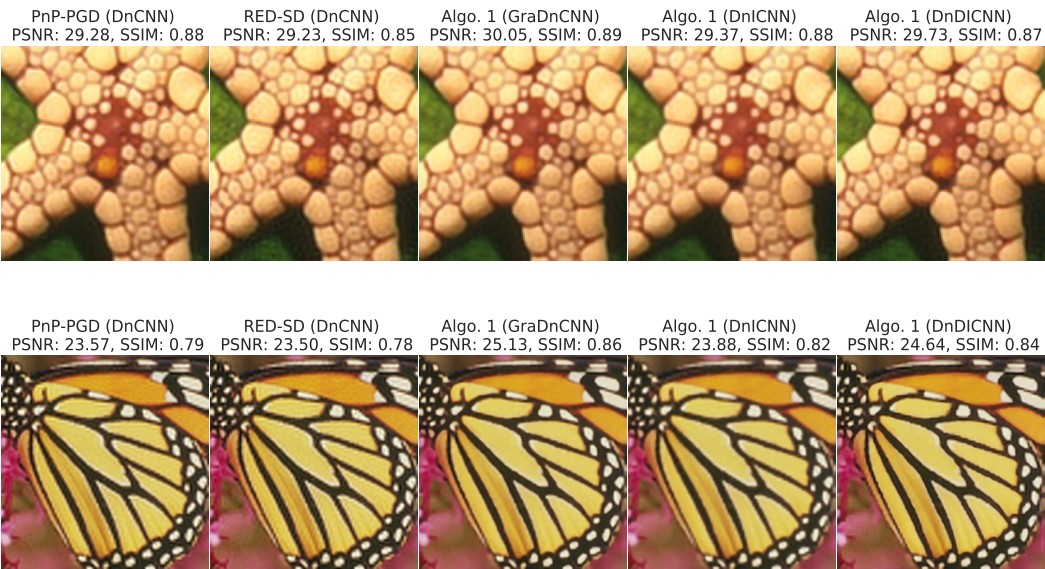

Figure 4: Recovery results of the tasks of (top) Gaussian deblurring and (bottom) super resolution.

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
