# Supplemental Material

## A  Direct Approach for Designing Gradient-Driven Networks

Here we aim to design a deep network which by construction is the gradient of a certain function. While achieving this goal is challenging in general, we next provide two conditions for a given network $D_\theta$ to be a gradient:

1. **Symmetric Jacobian**. Assume there exists a potential function $\rho(\cdot)$ such that $D_\theta = \nabla\rho$. Then, the network's Jacobian $J_D$ is actually the Hessian of $\rho(\cdot)$, hence, the Jacobian must be symmetric.

2. **Pseudo-Linear Form**. We require the network to be pseudo-linear such that $D_\theta(\mathbf{x}) = J_D(\mathbf{x})\mathbf{x}$.

The first property is a necessary condition for a network to be a gradient, and there is no apparent way to directly enforce it. Assuming the first condition is satisfied, then the second property is a sufficient condition which be fulfilled for example by using positive homogeneous activations and removing all the biases of the network, thus, making the entire network positive homogeneous ($D_\theta(c\mathbf{x}) = cD_\theta(\mathbf{x}), \ \forall c \geq 0$). Under these two conditions, it holds that $D_\theta(\mathbf{x}) = \nabla\rho(\mathbf{x})$ where

$$\rho(\mathbf{x}) = \frac{1}{2}\langle \mathbf{x}, \ D_\theta(\mathbf{x})\rangle.$$

## B  Descent Lemma

**Lemma 1.** *Let $F : \mathbb{R}^{d_x} \to \mathbb{R}$ be a continuously differentiable and smooth function with a Lipschitz constant $L$. Then, for any $\boldsymbol{x}, \boldsymbol{z} \in \mathbb{R}^{d_x}$*

$$F(\boldsymbol{z}) \leq F(\boldsymbol{x}) + \langle \nabla F(\boldsymbol{x}), \boldsymbol{z} - \boldsymbol{x}\rangle + \frac{L}{2}||\boldsymbol{z} - \boldsymbol{x}||_2^2$$

Thus, by setting $\mathbf{x} = \mathbf{x}_k$ and $\mathbf{z} = \mathbf{x}_{k+1}$ in the above inequality, it holds that for any $0 < \mu_k \leq \frac{1}{L}$

$$\begin{aligned}
F(\mathbf{x}_{k+1}) &\leq F(\mathbf{x}_k) - \mu_k||\nabla F(\mathbf{x}_k)||_2^2 + \mu_k^2\frac{L}{2}||\nabla F(\mathbf{x}_k)||_2^2 \\
&= F(\mathbf{x}_k) - \mu_k\Big(1 - \frac{\mu_k L}{2}\Big)||\nabla F(\mathbf{x}_k)||_2^2 \ \leq \ F(\mathbf{x}_k) - \frac{\mu_k}{2}||\nabla F(\mathbf{x}_k)||_2^2.
\end{aligned} \tag{20}$$

## C  Constrained Setup

One may consider additional regularization or constraints on the solution. For example, typically the recovered image values are restricted to lay in a desired range (e.g., $[0, 255]$). To that end, we extend our problem to the following formulation

$$\hat{\mathbf{x}} = \arg\min_{\mathbf{x}\in\mathbb{R}^{d_x}} f(\mathbf{x}) + \lambda g_\theta(\mathbf{x}) + h(\mathbf{x}), \tag{21}$$

where $h(\cdot)$ is a proper, convex and lower semi-continuous function. Note that the current objective function is neither differentiable nor smooth in general, thus it cannot be minimized by simple gradient descent. To overcome this difficulty, we apply proximal-type methods and replace the update rule in Algorithm 1 (lines 4 and 7) with

$$\mathbf{x}_{k+1} = \mathcal{P}_{\mu_k h}\bigg(\mathbf{x}_k - \mu_k\Big(\nabla f(\mathbf{x}_k) + \lambda\big(\mathbf{x}_k - \mathcal{D}_\theta(\mathbf{x}_k)\big)\Big)\bigg) \ \triangleq \ \mathcal{T}\mu_k(\mathbf{x}_k), \tag{22}$$

35th Conference on Neural Information Processing Systems (NeurIPS 2021).

where $\mathcal{P}_{\mu_k h}(\cdot)$ is the proximal operator of $\mu_k h(\cdot)$. The convergence analysis of Algorithm 1 with update rule (22) relies on the notion of a *gradient mapping*, defined as

$$\mathcal{G}_{\mu_k}(\mathbf{x}) \triangleq \frac{1}{\mu_k}\big(\mathbf{x} - \mathcal{T}\mu_k(\mathbf{x})\big), \tag{23}$$

The gradient mapping is the analogue of the gradient in the constrained optimization setup, since it holds that $\mathbf{x}_{k+1} = \mathbf{x}_k - \mu_k \mathcal{G}_{\mu_k}(\mathbf{x}_k)$ and it coincides with the gradient in the unconstrained case (i.e. when $h(\cdot) \equiv 0$). Furthermore, we show in the Supplemental that for any $0 < \mu_k \leq \frac{1}{L}$ the gradient mapping satisfies the following descent inequality

$$\tilde{F}(\mathbf{x}_{k+1}) \leq \tilde{F}(\mathbf{x}_k) - \frac{\mu_k}{2}||\mathcal{G}_{\mu_k}(\mathbf{x}_k)||_2^2, \tag{24}$$

where $\tilde{F}(\cdot)$ is the updated objective function of (21) and $L$ is the Lipschitz constant of the unconstrained objective function $F(\cdot)$. By following the same steps of the proof of Theorem 1 where we substitute $F(\cdot)$ and $\nabla F(\cdot)$ with $\tilde{F}(\cdot)$ and $\mathcal{G}_{\mu_k}(\cdot)$ respectively, we obtain that

$$\min_{m=0,1,\ldots,k-1} ||\mathcal{G}_{\mu_m}(\mathbf{x}_m)||_2^2 \leq \frac{2\max(L_0, \beta L)}{k}\big(\tilde{F}(\mathbf{x}_0) - \tilde{F}^*\big). \tag{25}$$

Thus, all limit points of the sequence created by Algorithm 1 with (22) are stationary points of $\tilde{F}(\cdot)$.

## D  Proofs

### D.1  Proof of Theorem 1

By inequality (14), for any $k \geq 0$ we have

$$||\nabla F(\mathbf{x}_k)||_2^2 \leq \frac{2}{\mu_k}\big(F(\mathbf{x}_k) - F(\mathbf{x}_{k+1})\big) \leq 2\max(L_0, \beta L)\big(F(\mathbf{x}_x) - F(\mathbf{x}_{x+1})\big),$$

where we use the the bound $1/\mu_k \leq \hat{L} \leq \max(L_0, \beta L)$. Summing the above inequality for $m = 0, \ldots, k-1$, we obtain

$$\sum_{m=0}^{k-1} ||\nabla F(\mathbf{x}_m)||_2^2 \leq 2\max(L_0, \beta L) \sum_{m=0}^{k-1}\big(F(\mathbf{x}_m) - F(\mathbf{x}_{m+1})\big) = 2\max(L_0, \beta L)\big(F(\mathbf{x}_0) - F(\mathbf{x}_k)\big).$$

Since $F^* \leq F(\mathbf{x}_k)$ for any $k \geq 0$, it holds that

$$\sum_{m=0}^{k-1} ||\nabla F(\mathbf{x}_m)||_2^2 \leq 2\max(L_0, \beta L)\big(F(\mathbf{x}_0) - F^*\big).$$

Therefore,

$$\min_{m=0,1,\ldots,k-1} ||\nabla F(\mathbf{x}_m)||_2^2 \leq \frac{1}{k}\sum_{m=0}^{k-1} ||\nabla F(\mathbf{x}_m)||_2^2 \leq \frac{2\max(L_0, \beta L)}{k}\big(F(\mathbf{x}_0) - F^*\big),$$

which completes the proof. $\qquad\square$

### D.2  Descent Inequality for Gradient Mappings

First, notice that we can rewrite (16) as

$$\mathbf{x}_{k+1} = \mathcal{P}_{\mu_k h}\Big(\mathbf{x}_k - \mu_k \nabla F(\mathbf{x}_k)\Big).$$

Thus, according to the properties of proximal operators [6], we have that

$$h(\mathbf{x}_{k+1}) \leq h(\mathbf{x}_k) + \langle \nabla F(\mathbf{x}_k), \mathbf{x}_k - \mathbf{x}_{k+1}\rangle - \frac{1}{\mu_k}||\mathbf{x}_k - \mathbf{x}_{k+1}||_2^2.$$

Recall that $\mathbf{x}_{k+1} = \mathbf{x}_k - \mu_k \mathcal{G}_{\mu_k}(\mathbf{x}_k)$, leading to

$$h(\mathbf{x}_{k+1}) \leq h(\mathbf{x}_k) + \mu_k \langle \nabla F(\mathbf{x}_k), \mathcal{G}_{\mu_k}(\mathbf{x}_k)\rangle - \mu_k ||\mathcal{G}_{\mu_k}(\mathbf{x}_k)||_2^2. \tag{26}$$

Furthermore, by Lemma 2 with respect to $F(\cdot)$, it holds that

$$F(\mathbf{x}_{k+1}) \leq F(\mathbf{x}_k) + \langle \nabla F(\mathbf{x}_k), \mathbf{x}_{k+1} - \mathbf{x}_k \rangle + \frac{L}{2} ||\mathbf{x}_{k+1} - \mathbf{x}_k||_2^2.$$

Rewriting the above using the gradient mapping, we obtain

$$F(\mathbf{x}_{k+1}) \leq F(\mathbf{x}_k) - \mu_k \langle \nabla F(\mathbf{x}_k), \mathcal{G}_{\mu_k}(\mathbf{x}_k) \rangle + \frac{\mu_k^2 L}{2} ||\mathcal{G}_{\mu_k}(\mathbf{x}_k)||_2^2. \tag{27}$$

Finally, note that the constrained objective function can be written as $\tilde{F} = F + h$. Hence, by summing inequalities (26) and (27), we get the following descent inequality

$$\tilde{F}(\mathbf{x}_{k+1}) \leq \tilde{F}(\mathbf{x}_k) - \mu_k \left(1 - \frac{\mu_k L}{2}\right) ||\mathcal{G}_{\mu_k}(\mathbf{x}_k)||_2^2 \leq \tilde{F}(\mathbf{x}_k) - \frac{\mu_k}{2} ||\mathcal{G}_{\mu_k}(\mathbf{x}_k)||_2^2,$$

for any $0 < \mu_k \leq \frac{1}{L}$.

Notice that for $\mathcal{G}_{\mu_k}(\mathbf{x}_k) \neq 0$ we get that $\tilde{F}(\mathbf{x}_{k+1}) < \tilde{F}(\mathbf{x}_k)$. Otherwise $\mathcal{G}_{\mu_k}(\mathbf{x}_k) = 0$ and we have

$$\mathbf{x}_k = \mathbf{x}_{k+1} = \mathcal{P}_{\mu_k h}\left(\mathbf{x}_k - \mu_k \nabla F(\mathbf{x}_k)\right).$$

This implies that

$$\mathbf{x}_k - \mu_k \nabla F(\mathbf{x}_k) - \mathbf{x}_k \in \mu_k \partial h(\mathbf{x}_k),$$

where $\partial h(\mathbf{x}_k)$ denotes the subdifferential of $h(\cdot)$ at $\mathbf{x}_k$. The above can rewritten as

$$0 \in \nabla F(\mathbf{x}_k) + \partial h(\mathbf{x}_k) = \partial \tilde{F}(\mathbf{x}_k),$$

indicating that $\mathbf{x}_k$ is a critical point of $\tilde{F}(\cdot)$.

# E    Additional Visual Results

Below we provide additional visual examples of Gaussian deblurring and super-resolution, shown in Fig. 4 and Fig. 5.

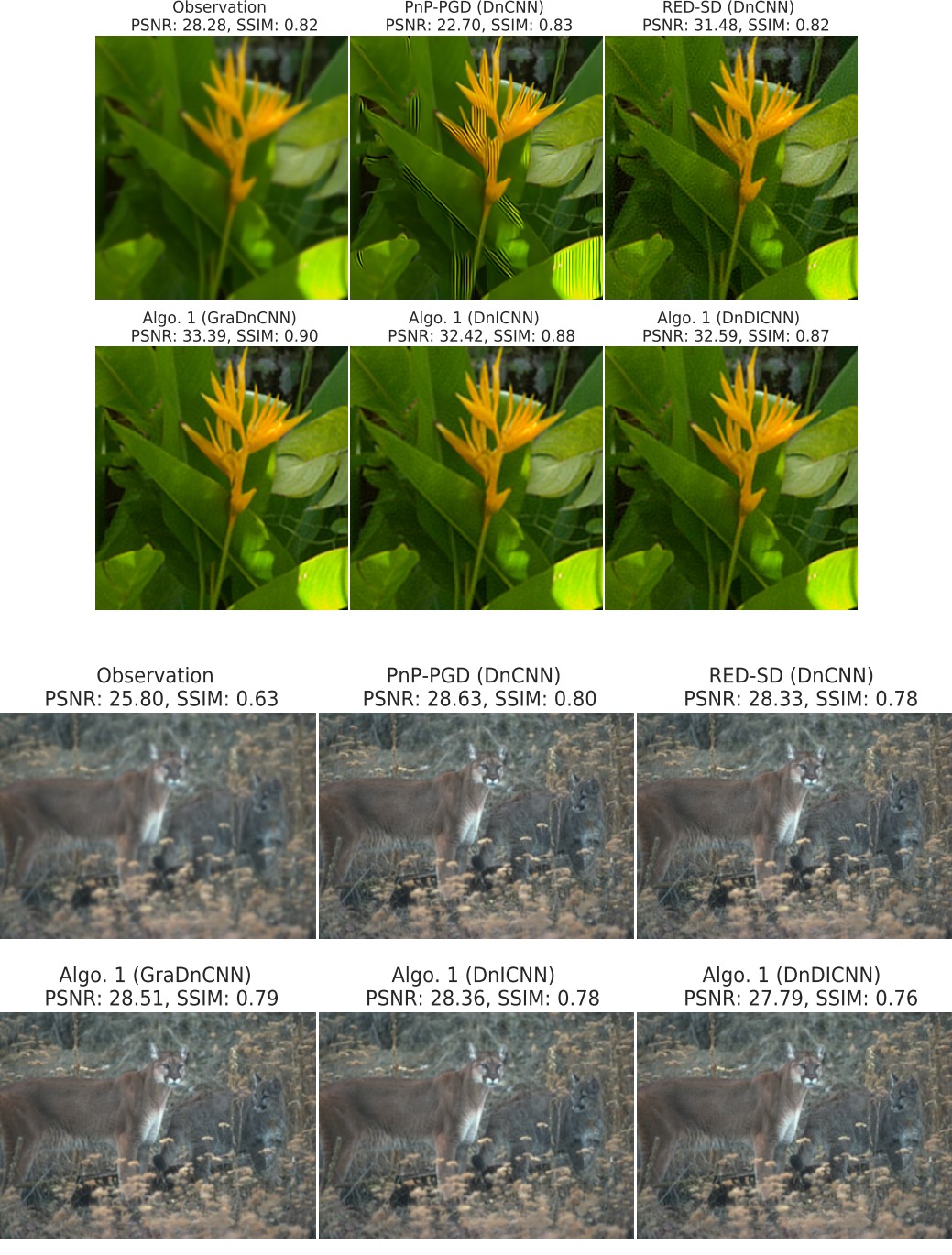

Figure 4: Recovery examples from (top) Set11 and (bottom) BSD500 for Gaussian deblurring.

# F  Hyperparameter Setting

To properly tune the various methods we performed an extensive grid search over the space of hyperparameters where the results are shown in Figures 6-9. For the initialization of the techniques we first test random and zero inputs. However, we obtained the best results when using the observed blurred images as initialization for Gaussian deblurring task, and using bilinearly upsampled versions of the input images as the initial solutions for single-image super resolution.

The selected hyperparameters, used in the experiments, are detailed in Table 2. Note that we add a projection step for both PnP-PGD and RED-SD to produce valid solution and improve stability, leading to the following update rules

$$\text{PnP-PGD:} \, \mathbf{x}_{k+1} = \mathcal{P}_{[0,255]}\Bigg( \mathcal{D}\Big( \mathbf{x}_k - \mu_k \nabla f(\mathbf{x}_k) \Big) \Bigg), \tag{28}$$

$$\text{RED-SD:} \, \mathbf{x}_{k+1} = \mathcal{P}_{[0,255]}\Bigg( \mathbf{x}_k - \mu_k \Big( \nabla f(\mathbf{x}_k) + \lambda \big( \mathbf{x}_k - \mathcal{D}(\mathbf{x}_k) \big) \Big) \Bigg). \tag{29}$$

Here $\mathcal{P}_{[0,255]}$ denotes the projection of image pixel values into the range $[0, 255]$.

Table 2: Parameter Setting

| Parameter | PnP-PGD | RED-SD | Algorithm 1 |
|:---:|:---:|:---:|:---:|
| Gaussian Deblurring | | | |
| $\lambda$ | - | 0.05 | 0.02 |
| $\mu_k$ | 1.99 | 1.99 | Backtracking |
| $L_0$ | - | - | $1 + \lambda$ |
| $\beta$ | - | - | $\sqrt{2}$ |
| Iterations | 125 | 200 | 400 |
| Super Resolution | | | |
| $\lambda$ | - | 0.05 | 0.04 |
| $\mu_k$ | 1.99 | 2.99 | Backtracking |
| $L_0$ | - | - | $1 + \lambda$ |
| $\beta$ | - | - | $\sqrt{2}$ |
| Iterations | 75 | 125 | 400 |

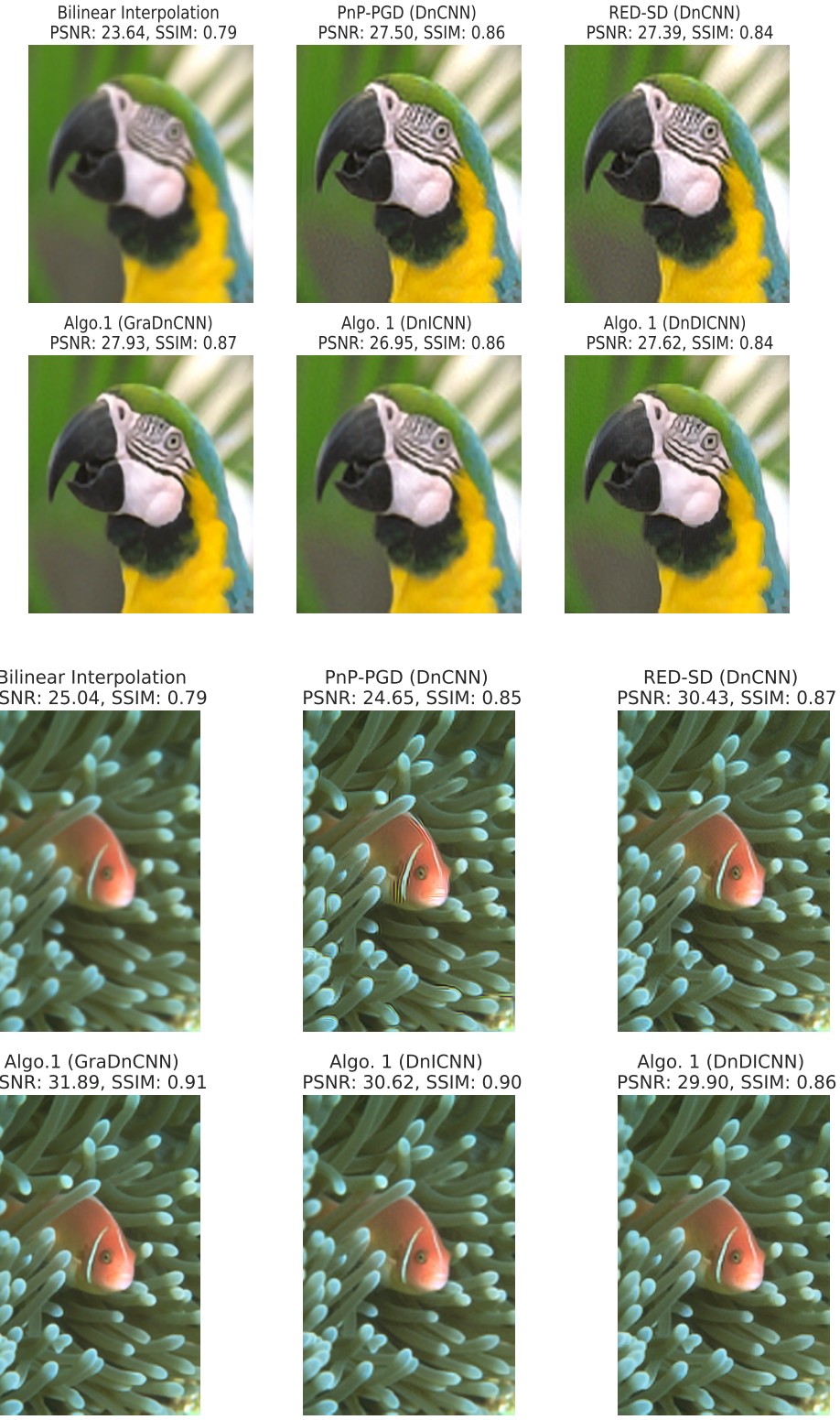

Figure 5: Recovery examples from (top) Set11 and (bottom) BSD500 for the task of super-resolution.

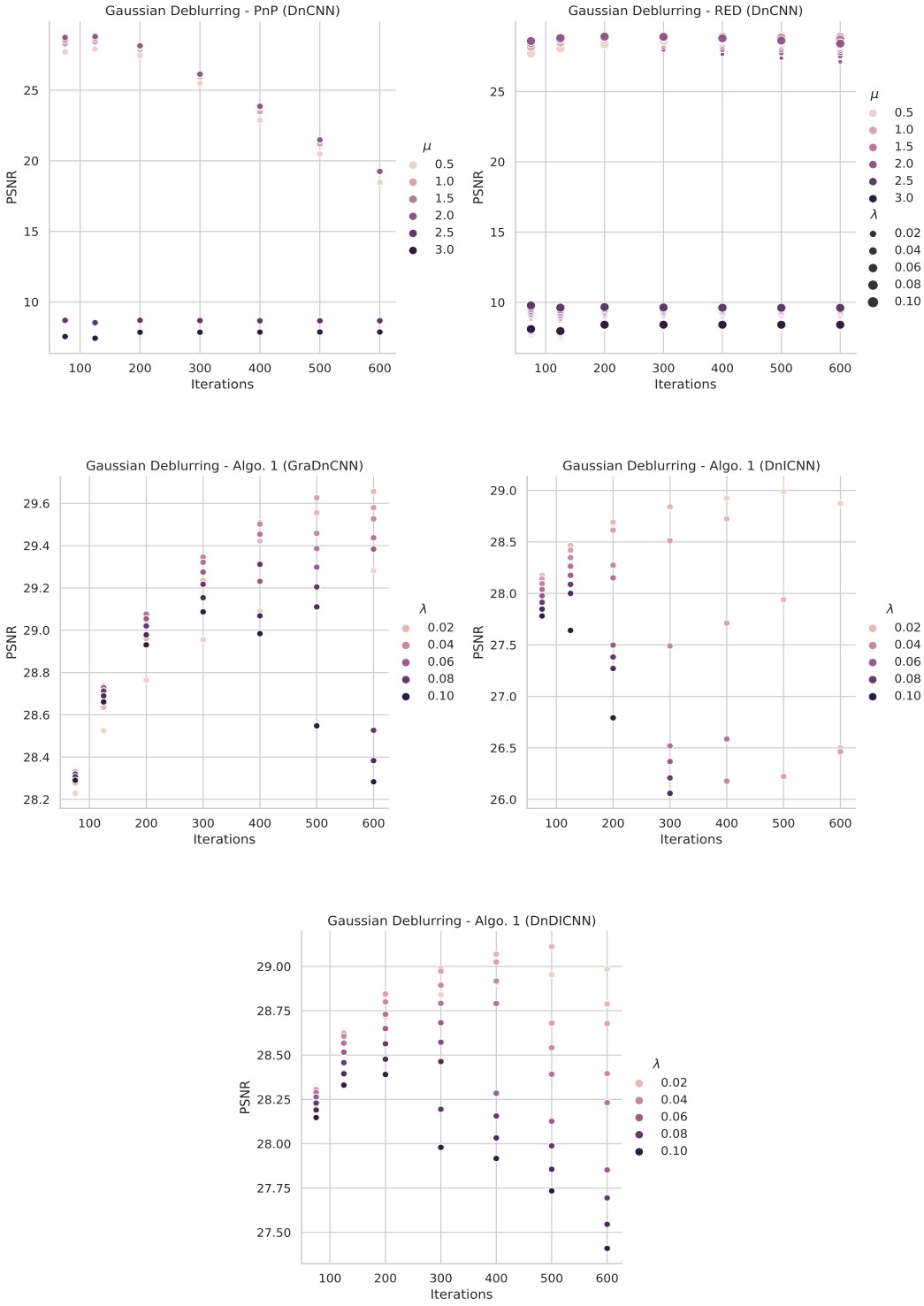

Figure 6: Hyperparameter Search - Gaussian Deblurring PSNR

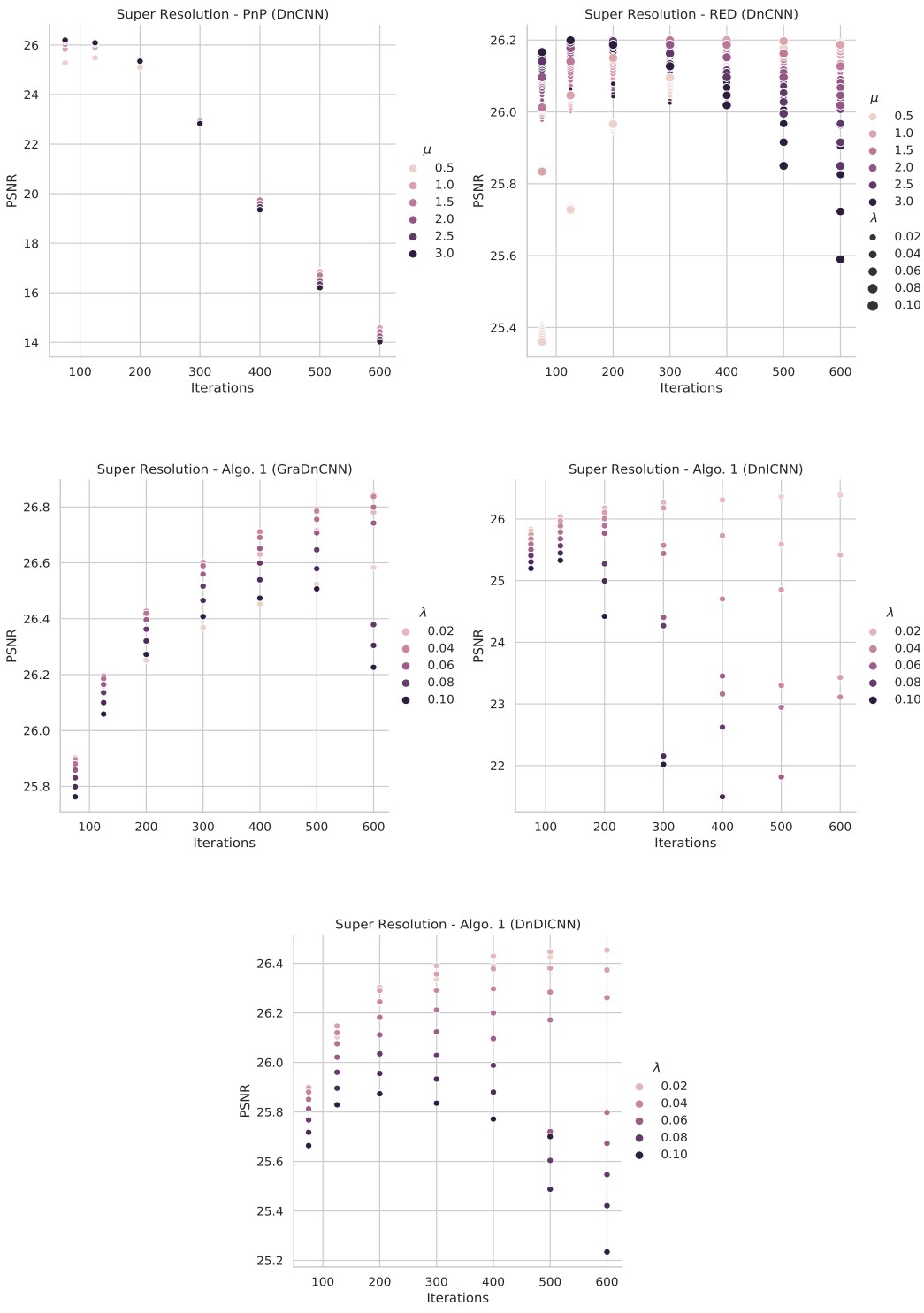

Figure 7: Hyperparameter Search - Super Resolution PSNR

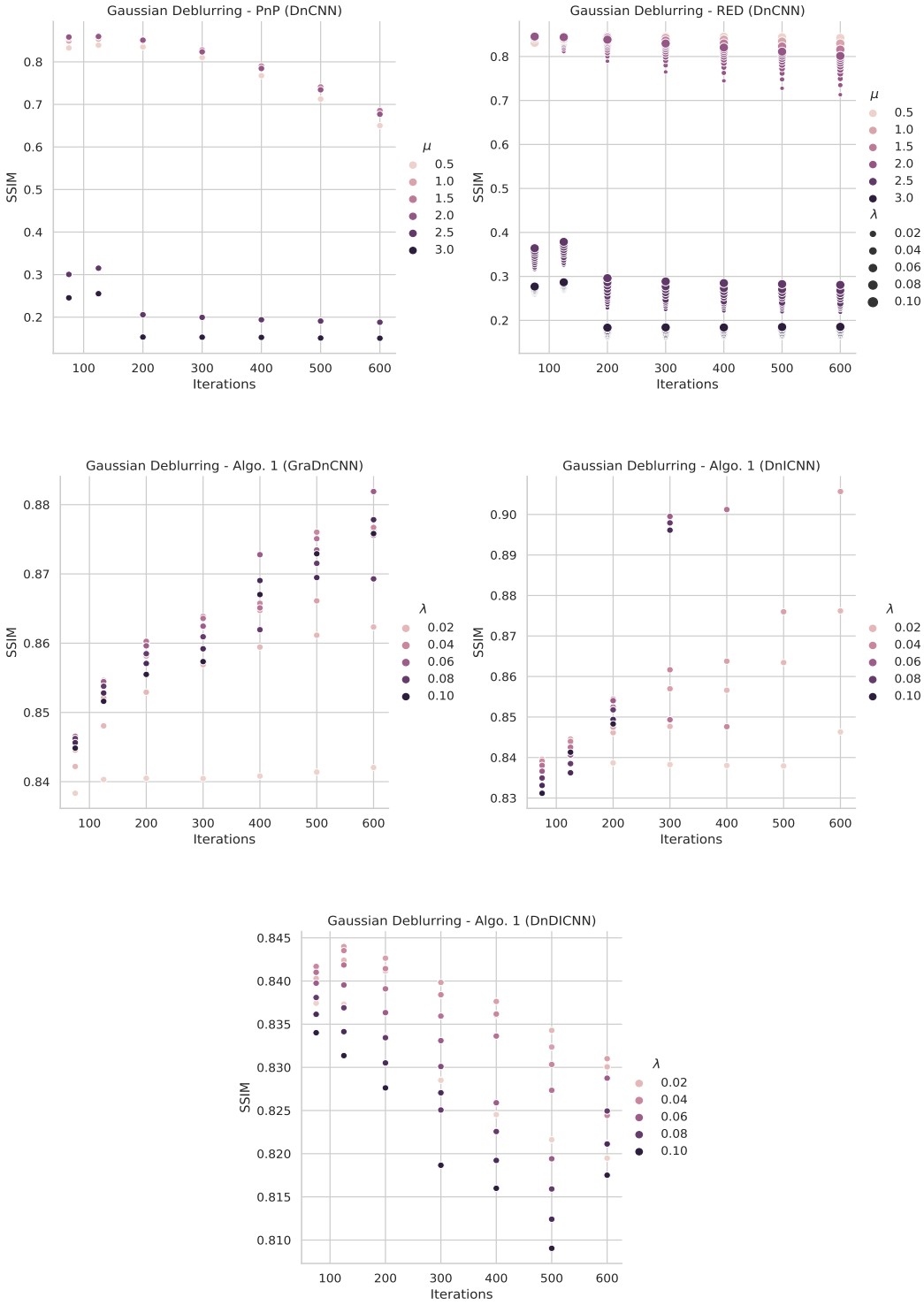

Figure 8: Hyperparameter Search - Gaussian Deblurring SSIM

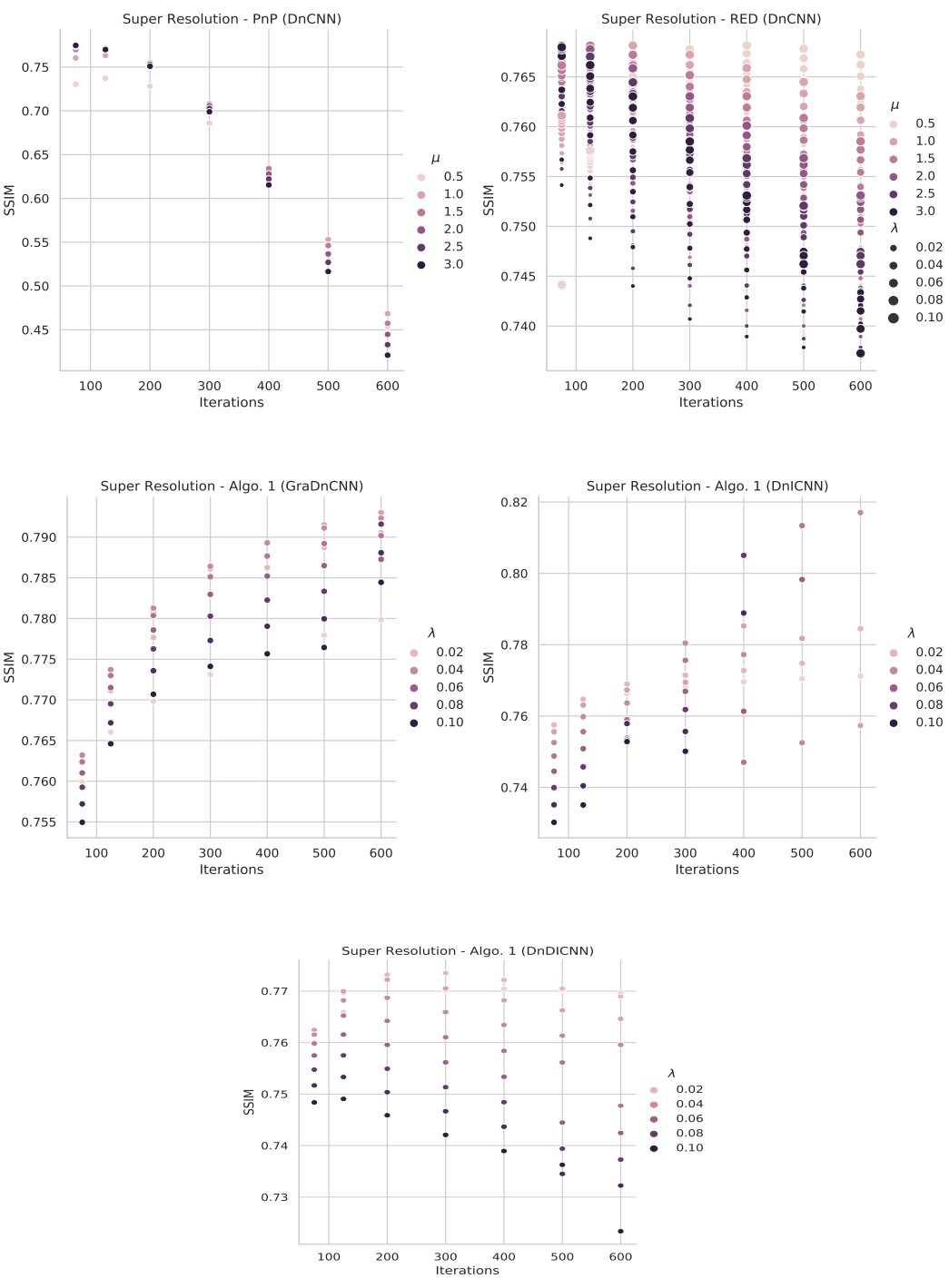

Figure 9: Hyperparameter Search - Super Resolution SSIM