# OpenReview forum: "It Has Potential: Gradient-Driven Denoisers for Convergent Solutions to Inverse Problems"
_NeurIPS.cc/2021/Conference — NeurIPS 2021 Poster_

### Official Review · Reviewer_rP8a · 2021-07-11

**Rating:** 7
**Confidence:** 4

**Summary:**

**Summary**:
This paper addresses an important issue in the framework of regularization by denoising (RED), which is that the explicit RED regularizer $\frac{1}{2}x^T(x-D_\sigma(x))$ exists only if a set of strict conditions on the denoiser are met. Due to this problem, RED fails to formulate explicit regularizers for advanced denoisers (e.g. CNN) since they generally do not satisfy the conditions. Although some work has established some theoretical analysis for RED algorithms using deep denoisers, a linkage between the denoiser and the regularizer is generally missing.

In this paper, the authors proposed to formulate the regularizer as a *smooth potential functional* $g(x)$, which is parameterized by a CNN with smooth activation functions. To link the potential functional to a denoiser, the authors trained the CNN such that the gradient $\nabla g(x)$ outputs the noise residual for the AWGN. In this way, the proposed method can formulate the gradient-based RED update by directly taking the gradient of the objective function $F(x)$.

Since the objective function is available, the authors also proposed to use the standard back-tracking trick in their framework to guarantee convergence. Additionally, a convergence analysis is also introduced by following the common nonconvex optimization analysis given the handy $F(x)$.

**Limitations And Societal Impact:**

**Weakness**

1. In the description under eq. (7), it would be clearer if the authors can explicitly link $\mathcal{W}$ and $\mathcal{V}$ to the convolutional operations.

2. Learning a denoiser as the gradient of a convex function is very interesting. But I do not get why having two convex potential $g_{\theta_1}$ and $g_{\theta_1}$ subtracting each other will increase the expressivity.

3. It is nice to have a back-tracking algorithm implemented in RED since the Lipschitz constant of a CNN is not easy to estimate. However, it seems to me that *Liu et al* has also proposed a back-tracking algorithm for RED. What is the difference between yours and theirs?

4. With all due respect and no intention to undermine the novelty, it is a little bit disappointing for me to see that there is only one theorem that only accounts for the nonconvex case. Given the fact that the authors also proposed a convex denoising functional, it may improve the completeness of the paper if the author can include another short theorem (even if it is well-known) or some remarks on the convergence of the proposed framework under convex $g(x)$.

5. In the experiments, the batch normalization and biases are removed for DnCNN, which may significantly reduce the representation power of the network. Can the authors explain this? On the contrary, the proposed network architecture has biases, which makes the comparison unfair.

6. Both PnP and RED achieve degraded performance compared with the results in other PnP/RED papers. I guess this is mainly due to the modified DnCNN architecture. The authors should compare their method with PnP/RED using standard DnCNN (with either batch normalization or biases). Moreover, please also compare against the Lipschitz-constrained (nonexpansive) DnCNN, which demonstrated good performance as well as convergence in many works.

7. Some missing citations related to RED, back-tracking, and learning functionals:

    [1] Sun et al Async-RED: A Provably Convergent Asynchronous Block Parallel Stochastic Method using Deep Denoising
    Priors. ICLR 2021

    [2] Liu et al RARE: Image Reconstruction Using Deep Priors Learned without Groundtruth. IEEE JSTSP 2020

    [3] Lunz et al Adversarial Regularizers in Inverse Problems NeurIPS 2018

**Main Review:**

**Originality**: Overall, this paper constitutes a solid contribution to the study of RED, proposing a new way to train a denoising prior and linking it back to a regularizer.

**quality & clarity**: The paper is well-written and easy to read. The background review of the related work is very thorough and comprehensive

**Significance**: I believe their contribution is of enough significance to the community.


**Time Spent Reviewing:**

1 hour for writing response. 2 hours for reading and evaluation.

---

> ### Author Response · Authors · 2021-08-10
> **Author Response to Reviewer rP8a**
>
> We appreciate the reviewer for providing valuable remarks and positive feedback.
>
> 1. The notations $W$ and $V$ under eq. (7) represent general linear operations, which we specifically set as convolutions in our proposed potential networks. Following the reviewer's remark,  we will clarify this connection in the revised manuscript.
>
> 2. Building convex potential networks in conjunction with the universal approximation theorem suggests that such networks can approximate any convex potential function. However, the true unknown potential we try to approximate may be non-convex. Hence, we extend our architecture to the difference of two convex functions which can approximate any function with bounded Hessian including non-convex mappings, thus improving the expressivity of the network.
>
> 3. First, please note that the back-tracking scheme shown in our paper is a common and well-known optimization technique, hence our main novelty is the use of this scheme together with our potential-driven denoisers. Second, the main difference between the back-tracking scheme in RARE and the one we present is that we rely on the existence of an explicit objective function. As proven in the paper, this ensures that there some constant $L$ such that for any $\gamma<\frac{1}{L}$ our algorithm converges and that the number of updates of the step-size is finite and bounded by $O(\log_2L)$. In the case of RARE, the back-tracking does not rely on an objective function, hence, the number of updates of the step-size theoretically may be infinite. To avoid that, the authors of RARE introduced an additional parameter $\rho$ and they stop the back-tracking for $\gamma<\rho$, however, this may be insufficient to ensure convergence in theory.
>
> 4. In the original version we did not include a corollary with respect to the convex case due to the lack of space. However, we accept the reviewer’s suggestion and we will incorporate a short theorem showing the convergence to global minima in case of convex potentials.
>
> 5. All networks presented in the experiments are without batch-normalization and biases, so the comparison is fair. We found that removing batch-normalization and biases does not affect the denoising performance or even improve it. This observation with respect to DnCNN is consistent with previous works (See [31] for example). However, we understand the reviewer’s concerns and we will include comparison with DnCNN with biases and batch-normalization in the revised manuscript (see summary of the results below).
>
> 6. Following the reviewer’s comments, we will add to the revised manuscript comparisons which include versions of DnCNN with biases, batch-normalization, and nonexpansive DnCNN trained with gradient penalty added to the training loss with different weights. Please see a summary of our results below.
>
> 7. We thank the reviewer for bringing these important references to our attention. We will add them in the revised manuscript.
>
> $\bf Result Analysis$ (PSNR/SSIM)
>
> $\bf Task: Gaussian Deblurring$
>
> $\underline{\text{DnCNN with bias}}$
>
>
> Set11-              RED:28.18/0.77, PnP:13.41/0.41
>
>
> BSD500-          RED:27.56/0.72,  PnP:14.77/0.51
>
>
>
> $\underline{\text{DnCNN with bias and batch-norm}}$
>
> Set11-             RED:28.41/0.79,        PnP:26.01/0.83
>
> BSD500-        RED:27.81/0.74,        PnP:27.06/0.83
>
> $\underline{\text{DnCNN with batch-norm}}$
>
> Set11-              RED:27.97/0.76,       PnP:23.9/0.79
>
> BSD500-          RED:27.48/0.72,       PnP:26.3/0.81
>
> $\underline{\text{DnCNN with gradient penalty (weight=1e-1)}}$
>
> Set11-            RED:27.92/0.76,       PnP:14.07/0.38
>
> BSD500-        RED:27.4/0.72,         PnP:13.55/0.31
>
> $\underline{\text{DnCNN with gradient penalty (weight=1e-3)}}$
>
> Set11-            RED:27.98/0.76,       PnP:14.04/0.44
>
> BSD500-        RED:27.34/0.71,         PnP:13.44/0.39
>
> $\underline{\text{DnCNN with gradient penalty (weight=1e-5)}}$
>
> Set11-            RED:28.09/0.77,       PnP:13.97/0.41
>
> BSD500-        RED: 27.38/0.71,         PnP:13.91/0.37
>
>
> $\bf Task: Super-Resolution$
>
> $\underline{\text{DnCNN with bias}}$
>
> Set11-            RED:26.09/0.76,       PnP:11.29/0.29
>
> BSD500-       RED:26.17/0.72,        PnP:11.81/0.35
>
> $\underline{\text{DnCNN with bias and batch-norm}}$
>
> Set11-            RED:26.42/0.78,       PnP:22.76/0.73
>
> BSD500-         RED:26.35/0.74,       PnP:23.98/0.72
>
> $\underline{\text{DnCNN with batch-norm}}$
>
> Set11-              RED:26.15/0.76,      PnP:19.6/0.63
>
> BSD500-          RED:26.21/0.73,      PnP:21.73/0.66
>
> $\underline{\text{DnCNN with gradient penalty (weight=1e-1)}}$
>
> Set11-            RED:26.03/0.75,       PnP:12.96/0.33
>
> BSD500-        RED:26.07/0.72,         PnP:12.76/0.28
>
> $\underline{\text{DnCNN with gradient penalty (weight=1e-3)}}$
>
> Set11-            RED:26.14/0.76,       PnP:12.75/0.35
>
> BSD500-        RED:26.07/0.71,         PnP:12.27/0.32
>
> $\underline{\text{DnCNN with gradient penalty (weight=1e-5)}}$
>
> Set11-            RED:25.86/0.75,       PnP:10.97/0.22
>
> BSD500-        RED: 25.92/0.7,         PnP:11.14/0.21

---

> > ### Comment · Reviewer_rP8a · 2021-08-29
> > **After rebuttal**
> >
> > Thanks for your comprehensive rebuttal.
> >
> > I am surprised that PnP performs so badly even with DnCNN using bias/batch norm. One possible explanation is that PnP diverges under these DnCNN denoisers. If this is true, it is better to include some explanation in the paper in order to avoid any misunderstandings. Other than that, the response addressed my concerns.

---

> > > ### Author Response · Authors · 2021-09-02
> > > **Author Response to Reviewer rP8a**
> > >
> > > We thank the reviewer for the important comments.  When performing PnP-PGD with DnCNN we have no guarantees for convergence, hence, the iterative process may diverge and lead to bad performance. As can be seen from Fig. 2, we can avoid divergence by performing early stopping for example. However, this way, similar to other common approaches to avoid divergence, requires manual tuning which varies from different tasks and datasets, making the inverse solver inflexible. For fair comparison and for the completeness of the paper, we will add the results of PnP-PGD performed with early stopping and discuss the advantages and drawbacks of such approaches. In addition, following the reviewer’s comments, we will extend our discussion on the convergence of PnP-PGD and add the explanation above.

---

### Official Review · Reviewer_fHLE · 2021-07-15

**Rating:** 5
**Confidence:** 5

**Summary:**

Motivated by the RED algorithm for solving inverse problems, the authors propose a neural-net denoiser that is the gradient of an explicit potential function.  This guarantees that the Jacobian is symmetric definite, which then guarantees that the RED algorithm minimizes an explicit cost function, which in turn makes convergence straightforward to prove, thereby solving an important open problem about the convergence of RED and plug-and-play (PnP)-type algorithms.  The fact that there is an explicit cost also enables the use of adaptive stepsize selection within RED, with one particular scheme proposed by the authors.  Numerical comparisons to RED and PnP algorithms show improved PSNR and SSIM performance, as well as enhanced stability.

**Ethical Concerns:**

The reviewer does not see any ethical issues.

**Limitations And Societal Impact:**

The reviewer does not see any potential negative societal impact.

**Main Review:**

I think the overall idea is great: design a neural-net denoiser that---by construction---equals the gradient of an explicit potential function.  To my knowledge, this is novel and it solves an important open problem about the convergence of PnP/RED methods.

But I fear that the numerical experiments are seriously flawed.  There are two reasons.
1. It's important to understand that the fixed points of PnP and RED algorithms are directly affected by stepsizes used in those algorithms (please see [3, p.108 and p.112]).  This stands in sharp contrast to convex optimization, where the stepsize does not affect the fixed point.  Thus, to show the true potential of PnP/RED, one must tune the stepsize.  This can be easily accomplished using the validation data.
2. There is a compounding problem, which has to do with differences among PnP/RED algorithms.  In the convex optimization setting, the PGD algorithm is unstable for large stepsizes, unlike the ADMM and primal-dual splitting (PDS) algorithms, where the stepsize can be chosen arbitrarily.  In the context of PnP, the PGD, ADMM, and PDS algorithms all have the same fixed points for a given stepsize (please see [3, p.111]), but PnP-PGD places severe limits on that stepsize, making many of those fixed points unstable.  In practice, the best stepsizes (e.g., in the sense of MSE or SSIM) are typically unstable for PnP-PGD, meaning that optimally tuned PnP-PGD is not competitive with optimally tuned PnP-ADMM and PnP-PDS.  Similar issues may arise with different RED algorithms.
In conclusion, if the goal is to compare to PnP algorithms, then PnP-PGD must be avoided and either PnP-ADMM or PnP-PDS must be properly tuned.  Likewise, the RED algorithm must be chosen wisely and tuned.  The experiments in the paper show PnP failing completely, but this is simply because the authors have not made a fair comparison to PnP.  Likewise, RED was not tuned and so it does not appear that the RED comparison is fair either.

Minor comments:
1. In the Related Work section, it may help to know that [3] is a comprehensive overview of PnP & RED methods and not just an application of PnP & RED to MRI.
2. In the Related Work section, for completeness, it may help to note that the AMP [7] and VAMP [17] frameworks are unique in guaranteeing that an MMSE denoiser yields and MMSE solution to the inverse problem, at least under large random forward operators.  In the same context, they prove convergence for Lipschitz denoisers.
3. Equation (13) is missing subscript k on x in two places.  Also, it should be noted immediately after (13) that (13) is RED-SD.
4. I suggest removing "It Has Potential" from the title.  You could add this phrase to any paper; it has no value.


**Time Spent Reviewing:**

3

---

> ### Author Response · Authors · 2021-08-10
> **Author Response to Reviewer fHLE**
>
> We thank the reviewer for the important comments and constructive criticism.
>
> 1. We completely agree with the reviewer’s remarks that the performance and stability of current common RED and PnP methods greatly depend on the tuning of the step size. This is a major drawback which typically requires cumbersome manual tweaking  for each specific problem setting or more sophisticated schemes such as in [47]. Therefore, one of the major goals of our work is remove this dependency of the solution on the step-size by introducing an alternative approach in which we have an automatic back-tracking scheme, enabled by potential-based denoisers. Incorporating these denoisers within RED and PnP frameworks will allow for the creation of methods, such as the one presented in the paper, that are more robust and require less tuning. Furthermore, we do not assume any available validation set, but only a proper model of the degradation operator and a trained denoiser.
>
> 2. First, we would like to emphasize that we do not aim to compete with PnP and RED frameworks, but rather demonstrate the advantages in incorporating potential-driven denoisers within these frameworks. Furthermore, we agree with the reviewer that there are other PnP algorithms that are more stable, hence, for fair comparison we will include in the revised manuscript results with a version of DnCNN for which PnP does not fail (see summary below). However, as mentioned by the reviewer, the solution (or performance) of such stable PnP methods is sensitive to the specific choice of the step size, hence, they require careful tuning of the step size. Thus, our work aims to provide an alternative approach that allows for a simple automatic tuning of the step-size, ensuring both convergence and that the final solution will not depend on the choice of the step size. Following the reviewer’s comments, we will emphasize the points mentioned above in the revised manuscript.
>
>
> Response to minor comments:
>
> 1. We will clarify this in the revised manuscript.
> 2. We appreciate the reviewer for bringing this to our attention. We will include these important remarks in our revised manuscript.
> 3. Thank you for pointing this out. We will correct it in our revision.
> 4. The phrase “It Has Potential” is given to suggest/hint that the proposed denoiser have potential functions from which they are derived. However, we value the reviewer’s opinion and we will consider removing it.

---

> > ### Comment · Reviewer_fHLE · 2021-08-23
> > **Reply to authors' comments**
> >
> > Thanks for your response.
> >
> > I don't agree with the comment "we do not aim to compete with PnP and RED frameworks."  Those frameworks, as originally proposed, are based on the idea that one can plug in *any* denoiser but give no guarantee of convergence.  Your approach requires one to use a constrained denoiser that guarantees convergence but that may have reduced performance due to the constraints imposed upon it.  The key question is: how much performance do you give up by imposing these constraints?  The current simulations do not answer this question.
> >
> > Also, I agree with the other reviewers that the paper could benefit from less discussion of convergence and more discussion on the network constraints needed for the denoiser to yield a valid gradient.

---

> > > ### Author Response · Authors · 2021-08-26
> > > **Author Response to Reviewer fHLE**
> > >
> > > We agree with reviewer that any denoiser can be plugged into PnP and RED frameworks but with no guarantee of convergence.
> > > Ensuring convergence is not a theoretical issue, but it is important from a practical point of view since in many applications (e.g. medical imaging) stability plays a crucial role and one would like to ensure a stable recovery regardless of the data being used and less manual parameter tuning as possible. For this reason, a vast number of recent works (such as [51], [10], [32], [38], [42]) focus on studying the convergence of RED and PnP methods and providing specific conditions/constraints (e.g. non-expansiveness/Lipschitz-1) on the denoiser in-hand to ensure convergence.  We continue this line of research and present the notion of potential (or gradient) driven denoisers which can be plugged into RED and PnP methods and ensure stable recovery. Compared to previous works,  we impose the constraints on the denoiser directly through its aracitrucre, rather than performing spectral normalization or adding gradient penalty during training for example.
> > > To build such denoiser, we design a deep network $g_\theta(x):\mathbb{R}^n\rightarrow\mathbb{R}$, compute its gradient w.t.r to the input $x$ and model our denoiser based on this gradient $D(x)=x-\nabla_x g_\theta(x)$. Hence, any scalar-valued deep network can be used as our potential $g_\theta(x)$ to yield a valid gradient. In case we want to ensure convergence/stability when the denoiser is plugged into RED or PnP, it is sufficient to require the potential network to be a smooth function, implying that for in standard CNN the activations should be smooth functions such as ELU (which we use in our paper). This use of a smooth potential network leads to the denoiser GraDnCNN.
> > > Similar to previously proposed constrained denoisers (e.g. nonexpansive denoisers), we show (see below) that plugging GraDnCNN into an inverse optimization solver leads to a stable (provably convergent) method and to comparable or improved recovery performance in comparison with DnCNN for the tasks of super-resolution and deblurring. This implies that although we pose constraints on our denoiser, it does not lead to reduced recovery performance, but provides stable solution.
> > >
> > > Following the reviewer's comments, we will include the important points discussed above in the main body of our paper, and will add experimental results comparing our denoisers with DnCNN in terms of denoising performance.
> > >
> > > $\underline{Deblurring}$
> > >
> > > DnCNN (with batch norm)
> > >
> > > Set11 -     RED: 28.41/0.79,    PnP:26.01/0.83
> > >
> > > BSD500 - RED: 27.81/0.74,    PnP:27.06/0.83
> > >
> > > GraDnCNN
> > >
> > > Set11 -     Algorithm 1: 29.42/0,86.
> > >
> > > BSD500 - Algorithm 1: 28.91/0.83.
> > >
> > >
> > > $\underline{Super-Resolution}$
> > >
> > > DnCNN (with batch norm)
> > >
> > > Set11 -     RED: 26.42/0.78,     PnP:22.76/0.73
> > >
> > > BSD500 - RED: 26.35/0.74,     PnP:23.98/0.72
> > >
> > > GraDnCNN
> > >
> > > Set11 -     Algorithm 1: 26.69/0.79.
> > >
> > > BSD500 - Algorithm 1: 26.57/0.74.

---

### Official Review · Reviewer_UqjB · 2021-07-15

**Rating:** 7
**Confidence:** 4

**Summary:**

The paper proposes image denoising based CNNs as regularizers that can be constructed as the gradient of smooth potential functions $g$. The main promise is mentioned to be making an optimization algorithm has explicit regularization, when equipped with those denoisers. Unlike original regularization by denoising (RED) that the explicit regularizer exists if the denoisers satisfy several strict conditions, the proposed denoisers only rely on smoothness. Particularly, the denoisers consist of smooth activation functions to ensure the differential continuity and are trained in a residual denoising fashion using MSE loss. As a result, the proposed algorithm can be formulated as steepest descent variant of RED optimized by directly adopting the gradient of the objective function $F$. Standard back-tracking is used to ensure convergence. A convergence analysis is provided following the common optimization theory. Finally, the performance of the proposed method is evaluated over two image inverse problems such as gaussian deblur and super-resolution, with satisfactory results compared to existing methods based on PnP and RED.

**Limitations And Societal Impact:**

**Weakness:**

- Using the assumption that the denoiser $D$ is the proximal operator of some convex scaler function is to be fine to obtain some fundamental insights for the link between the CNN denoiser and regularization, but to me, it also relies on the convexity of the denoiser. Considering that Algorithm 1 equipped with non-convex denoiser achieves the best performance, it seems to be tricky to follow _Lemma 1_ to design the desnoisers.

- For PnP-PGD, RED-SD and the proposed method with GraDnCNN using DnCNN are trained without lipschitz constrain for $L < 1$, resulting a non-convex problem. Hence an initial point of the update sometimes is important. It would be interesting to see how different initialization influence the final reconstruction results.

- Similar to the above, both PnP and RED led to poor results compared to most of existing works. This may be due to the improper parameter tunning such as regularization trade-off $\lambda$, step-size $\gamma$ and denoising intensity.
A recent work by (Xu’2021) introduce a simply way enable the tuning of the regularization parameter within PnP, it is not clear for me why this easy approach is not applied for PnP.


**Main Review:**

**Originality:** This paper is well motivated and is written fairly well. The idea sounds simple but promising and makes it possible to be directly applied for other variant RED algorithms. Most importantly, this paper provides a new guide of training CNN denoisers as explicit regularizers, unlike current literature, where the link between denoisers and regularization is commonly missing.

**Quality & Clarity:**  Overall the paper is  easy to read. The background review of the related work is comprehensive and clearly stated. However, I found it hard to follow at some specific sections due to insufficient details on the notations.  For one, it is a little bit confusing for me about the function $h$ mentioned in Lemma 1, line167 -- 172 and the one in Eq. 15.

**Significance:** I think this is a strong paper, in a line of research that is interesting and should be of interest to the NeurIPS community.


**Time Spent Reviewing:**

3

---

> ### Author Response · Authors · 2021-08-10
> **Author Response to Reviewer UqjB**
>
> We would like to thank the reviewer for the valuable comments and positive feedback.
>
> Clarity:
>
> We appreciate the reviewer’s remarks. We will improve the clarity of the revised paper by  proofreading the manuscript and clarifying the notations where needed, as in the sections mentioned by the reviewer.
>
> Weaknesses:
>
> 1. We understand the reviewer’s concerns. The proximal operator is a fundamental form of a denoiser which has been used for decades in the area of inverse problems. Hence, we use Lemma 1 as the inspiration and the motivation for our work. However, the rise of deep learning in recent years has proven the benefits of non-linear and non-convex optimization.  Hence, while our starting point is convex optimization, our work extends beyond it and we show that even in the general case where our potential function is not convex, we can still obtain both stable and high-performance recovery. Thus, in the context of inverse problems, our work aims to bridge the gap between well-known theory of convex optimization and the highly non-linear non-convex nature of deep neural networks.
>
>
> 2. We did experiment with different initializations, however, we did not include the analysis in the original paper due to the lack of space. In our experimented we test three different initializations:
>
> A) Zeros.
>
> B) Random.
>
> C) We use the observed image (measurements) for the deblurring task, and bilinearly upscaled version of the observed image for the super-resolution task.
>
> We found that the third initialization leads to the best results for all methods, thus, we use it in the experiments shown in the original submission. However, following the reviewer’s comments, we will include in the revised manuscript an analysis of the recovery results for different random initializations.
>
> 3. We agree with the reviewer that intensive parameter tuning would probably lead to better performance. However, this typically requires cumbersome manual tweaking  for each specific problem setting or more sophisticated schemes such as the one shown in [47] that requires reinforcement learning. Therefore, one of the goals of our work is to significantly reduce the need for parameter tuning by using an automatic back-tracking step-size, enabled by the existence of a potential function of our denoiser. Thus, both the stability (convergence) and the nature of our final solution do not depend on the specific choice of the step-size as in common PnP and RED methods. Finally, we would like to emphasize that we do not aim to compete with PnP and RED frameworks, but rather demonstrate the benefits in incorporating potential-driven denoisers into these frameworks.

---

### Official Review · Reviewer_2weA · 2021-07-16

**Rating:** 6
**Confidence:** 4

**Summary:**

Summary:
- This submission deals with designing stable and smooth denoisers that could be directly used for regularization, or for plug-and-play (PnP) methods to solve inverse problems.
- This is an improvement upon existing denoisers such as RED and PnP which are not necessarily stable.
- The key idea is to craft neural networks that output scalar-valued potential function g whose gradient is the denoiser residual. To guarantee convergence and stability, g has to be smooth and h=1/2x^2 - g convex.
- Advantages: the denoisers lead to symmetric Jacobians, allowing for a MAP estimation; denoisers can be integrated into optimization schemes with backtracking step size, and thus no need for enforcing Lipschitz continuity.
- To construct such a network, GraDnCNN stacks conv layers and nonlinear activations, and proposes DnICNNs with non-negative weights and convex activations. Variants of DnICNN are discussed that achieve better denoising/recovery performance for Gaussian deblurring, and image superresolution.

Contributions:
- construct stable denoisers based on neural networks as gradients of smooth potential functions g
- integrate the denoisers as a regularizer for solving inverse problems and test it out for gaussian image deblurring and superresolution which outperforms RED and traditional PnP methods


**Limitations And Societal Impact:**



**Main Review:**

Originality and significance
- a novel perspective to craft CNNs for designing stable denoiser that leverage the power of neural nets in a guaranteed manner
- The design seems to be limiting, needing non-negative weights, and thus the scope and usefulness of this theory is not very clear. This is discussed in the comments next.

Clarity:
- This paper is crowded, and it discusses different aspects of the potential based denosiers, so the main novelty of this work is lost.
- It could benefit from a better organization. More details about the architecture of ICNN and requirements on the weights would help better understand the paper. This reviewer doesnt think the discussions about the convergence are of high priority as those come natural given the constraints of convexity and smoothness on the denoiser.

Comments:
- This is an Interesting connection between convex regularization and CNNs. How far the network design can go? Let’s say we have off-the-shelf deep CNN based denoisers such as U-net and want to see their corresponding potential function, and regularization function.  It's important for the theory to be insightful and suggest good architectures. The Author's comment would be helpful.

- About the GDCNN, what restications on the weights and layers are required to make sure the output is a legitimate gradient for a lipschitz and convex denoiser? It’s not clear that the non-convex training objective in (9) would naturally lead to network weights that would produce legitimate gradients?

- The constraints on the design of DnICNN seem to be pretty stringent; non-negative weights and convex activations. It’s strange that no weight normalization is needed to ensure lipschitzness. Is that correct?

- If I understand correctly, only ICNN is convex. DnDICNN and GraDnCNN are not legitimate for stability. right?

- in order to see the usefulness of the connection between regularization and neural nets, it is useful to include off-the-shelf denoisers in the experiments Fig. 2,3. One such example is unrolled networks that are trained end-to-end, or the inverse problem with the generative model projection. Glancing through the literature, a few examples, very relevant to this work include [1], [2].


[1] Mardani, M., Sun, Q., Papyan, V., Monajemi, H., Vasanawala, S., Pauly, J., Donoho, D. (2018). Neural Proximal Gradient Descent for Compressive Imaging. Advances in Neural Information Processing Systems, 31, 9573-9583.

[2] Van Veen D, Jalal A, Soltanolkotabi M, Price E, Vishwanath S, Dimakis AG. Compressed sensing with deep image prior and learned regularization. arXiv preprint arXiv:1806.06438. 2018 Jun 17.



**Time Spent Reviewing:**

---

> ### Author Response · Authors · 2021-08-10
> **Author Response to Reviewer 2weA**
>
> We appreciate the reviewer for the time and effort spent in providing us with these important comments and constructive feedback.
>
> Originality and significance:
>
> We agree with the reviewer that the use of non-negative weights may be limiting. However, please notice that this restriction holds only for DnICNN, while in the more general case we use  GraDnCNN which has no restrictions on its weights. Thus GraDnCNN can be easily trained and applied within our inverse scheme, leading to probable convergence.
>
> Clarity:
>
> We understand the reviewer’s concerns. Following the comments, we will remove certain standard optimization results and replace them with appropriate citations, thereby saving space to provide more details on the architecture of our potential networks.
>
> Specific responses:
>
> 1. In general we want the trained denoiser to be the gradient of an explicit potential function which we can evaluate. To achieve this, we can use the following approaches:
>
> A) Directly designing a denoiser which is a gradient of a potential function. This approach first requires the Jacobian of the denoiser to be symmetric (since a gradient must have symmetric Jacobian). In addition to the previous condition, an additional sufficient condition is the denoiser $D$ is a locally homogeneous function, i.e., $D(cx)=cD(x)$ where $c$ is a scalar such that $|c-1|<=\epsilon$ for some small $\epsilon$. One simple way to achieve this is to remove all biases from the denoising network, making it globally homogeneous.
>
> Assuming the two conditions above hold, we can show that $D$ is the gradient of a potential function $g$ given by $g(x)=\frac{1}{2}\langle x,D(x)\rangle$. Hence, the network design can be quite general.
>
> B) Notice that most common denoisers, such as DnCNN and UNet, do not have symmetric Jacobian, thus, they do not satisfy the conditions mentioned above. Enforcing these conditions may be challenging, hence, we provide in our paper a simple alternative in which we explicitly construct the potential function as a deep network and take the denoiser to be its gradient. The potential network can have almost any architecture as long as it has smooth activations and its output is a scalar.
>
> 2. Please note that we build our potential function as the deep networks and the denoiser is the gradient of this network. Hence, the denoiser is a gradient by construction and we enforce the conditions on the potential network. In the case of GraDnCNN, we have no restrictions on the weights of the potential networks. The only conditions are that the activations are smooth functions and the output of the potential network is a scalar.
>
> Once we have our potential network, we can use its gradient as our denoiser and train using the training objective in (9) which is a standard (non-convex) loss function for performing residual learning, typically used for training denoisers such as DnCNN.
>
> Only in the case of  DnICNN where the potential function is convex restrict the weights to be non-negative. Following the reviewer’s comments, we will provide detailed explanations and clarifications about the network architecture in the revised manuscript.
>
> 3. Weight normalization is typically used to bound the Lipschitz constant of the resultant network. However, we intentionally avoid this approach of controlling the Lipschitz constant and provide an alternative by using a back-tracking step-size, enabled by our potential network, which does not require the knowledge of the Lipschitz constant (or bounding it). The use of smooth activation functions guarantees that our gradient-based denoiser is Lipschitz for some unknown constant $L$.
>
>
> 4. All presented denoisers (DnICNN, DnDICNN, GraDnCNN) allow stable recovery with guaranteed convergence. The difference is that the use of DnICNN in our optimization schemes leads to convergence to global minima while the DnDICNN, GraDnCNN provide convergence to local minima in general.
>
>
> 5. In our experiments, we use DnCNN as a deep learning reference, which is a common and widely-used denoiser. We thank the reviewer for providing citations [1] and [2] which are important in the context of regularization using deep neural networks, and we will discuss them in the revised manuscript. However, as we state at the end of Section 2, we do not aim to compete with end-to-end learned solutions for specific inverse problems such as [1] or [2], but rather we wish to train a denoiser one-time and use it for all inverse problems without additional training (given appropriate models for the degradation processes).

---

> > ### Comment · Reviewer_2weA · 2021-08-30
> >
> > Thanks for the comments. I understand now that all four algs would be convergent, but not necessarily to global optimum. It’s also clear now that Lipschitz constraint is also satisfied with the backtracking. Thus, I tend to raise my score to 6.
> >
> > However, reading other reviewer’s comments and checking the simulation results in more details, for a fair comparison, the results in Fig. 2 need to be tuned well for a good range of hyperparameters, in terms of learning rate and lambda. Also, a larger number of iterations should be displayed to see the convergence. It is also unclear why DnICNN alg, which is the convex one with positive weighting (and thus more restrictive), converges much faster and leads to smaller mse. These need to be clarified in the paper.

---

> > > ### Author Response · Authors · 2021-09-02
> > > **Author Response to Reviewer 2weA**
> > >
> > > We would like to thank the reviewer for the positive feedback. Following the reviewer’s suggestions, we will incorporate into the revised manuscript a significantly more extensive analysis of our experiments, including the results produced for this rebuttal as well as a thorough study of the different hyperparameters.
> > >
> > > As for DnICNN, we agree with the reviewer that DnICNN is more restrictive than e.g. standard DnCNN, hence, we understand why it may be surprising that DnICNN lead to better performance. However, the use of DnICNN and its convex potential networks lead to inverse problems with a well-defined objective/loss function which is particularly smooth and convex. This implies that the loss landscape/structure is well-behaved, thus, facilitating the optimization process. Furthermore, the fact that DnICNN is a less powerful denoiser than DnCNN may not necessarily be a disadvantage in iterative optimization processes where we aim for stability. This can be explained via an analogy between the denoising power in our inverse solvers and the learning rates (or step size) in general gradient-based optimization  - typically one would aim for using large learning rates ($\sim$powerful denoisers) to expedite the optimization process, however, too large learning rates may increase the loss and even lead to divergence. By contrast, small learning rates ($\sim$weaker denoisers) ensure stability and convergence, although they may seem less effective in minimizing the loss.

---

### Official Review · Reviewer_Lp9c · 2021-07-19

**Rating:** 5
**Confidence:** 5

**Summary:**


This paper considers the problem of designing an image denoiser that can be used as a prior within plug-and-play priors (PnP) or regularization by denosing (RED). Prior work has proved the convergence of PnP/RED for contractive, nonexpansive, and/or firmly nonexpansive denoisers. The design of denoisers that enable PnP/RED convergence is an active research area.

This paper focuses on denoisers that correspond to the gradients of some potential functions. By using such denoisers within RED, one can guarantee convergence via traditional optimization. The key idea in the paper is to use a deep neural network to parameterize a potential function (instead of the denoiser), then train that function for denoising by using its gradient. Additionally, when the weights of the network are nonlinear and activation functions are nondecreasing and convex, the denoiser corresponds to a convex function.

The key contribution of the paper is the formulation of a procedure for training a potential function that can be used to obtain a denoiser. The theoretical arguments presented in Lemma 2 and Theorem 1, are well known in optimization (see below for reference). Similarly, Algorithm 1 corresponds exactly to the RED-SD algorithm using a potential-driven denoiser. The empirical results compare three proposed variants of denoisers against PnP-PGD and RED-SD on problems of image deblurring and superresolution.

**Ethical Concerns:**

No issues

**Limitations And Societal Impact:**

No issues

**Main Review:**

Strengths: The central idea in the paper nicely complements the original work on RED, which attempted to obtain a PnP algorithm minimizing an explicit objective function. The recent literature on PnP/RED has focused on using deep neural nets to parameterize the denoiser. This paper argues that by instead parameterizing the potential function, one can obtain an explicit objective under milder assumptions than in the original RED paper.

Issues: (a) The first major issue in this publication is the discussion in “Section 3.2 Convergent Inversion Method”. The theoretical results claimed in that section in Lemma 2 and Theorem 1 are standard results from optimization (see, for example, “Section 1.2.3 Gradient method” in Nesterov's book “An Introductory Lectures on Convex Optimization”). One can simply cut these results by including references and save 1 page for providing other details. (b) The second major issue in the paper is in inadequate simulations. The paper seems to want to show that the proposed denoiser leads to (i) convergence; (ii) better solutions in terms of PSNR/SSIM than PnP/RED with other denoisers. Regarding (i): (1) There are a number of variants of PnP/RED that achieve provable convergence using nonexpansive variants of DnCNN (see denoiser in [40]). Why would you pick the one that makes PnP-PGM diverge? (2) Having a convex optimization reference, such as FISTA-TV, would help to understand the benefit of having the potential function. (3) Having convergence plots in terms of PSNR (dB) would help to see the speed of improvement in imaging quality. (4) Having a basic deep learning reference such as UNet would help to evaluate that all methods are properly implemented and take advantage of the measurement model. (5) Having a convergence plot in terms of the objective function, would help to validate that the GraDnCNN is indeed minimizing an objective. Regarding (ii): Why would we expect the existence of the potential function to lead to better imaging quality?

Summary: This paper is proposing an intriguing alternative to training nonexpansive denoisers for PnP/RED. While “it has potential”, the paper falls short in terms of fully convincing the benefit of the proposed method for solving inverse problems. Specifically, experimental validation is incomplete and needs to be significantly improved following my suggestions above. Additionally, the paper claims a number of classical results from optimization as contributions.

**Time Spent Reviewing:**

3 hours

---

> ### Author Response · Authors · 2021-08-10
> **Author Response to Reviewer Lp9c**
>
> We would like to thank the reviewer for the careful assessment of our paper and for providing valuable feedback.
>
> a. We would like to emphasize that we do not claim the standard optimization results are our contributions, but rather we simply provide them for the completeness of the paper and utilize them to prove convergence in the context of our potential-based denoisers. However, we understand the reviewer’s concerns, hence, we will replace certain theoretical derivations with proper references to save space in the revised manuscript.
>
> b.  Convergence :
> 	1) We agree with the reviewer that for nonexpansive denoisers, as the one in [4],  the PnP algorithms typically converge.  However, there are two main issues with nonexpansive denoisers. First, to the best of our knowledge, there is no practical way to prove that a given CNN denoiser is indeed nonexpansive, and this property is typically verified only empirically. Thus the given denoiser may be expansive for certain unknown inputs. Second, precisely enforcing this constraint is impractical, therefore, it is typically approximated by using methods such as spectral normalization which may be overly restrictive [see Kligvasser 2021,Sparsity Aware Normalization for GANs]. Therefore, we present in our paper an alternative approach which does not require controlling the denoiser’s Lipschitz constant, facilitating the training process of the denoiser. Moreover, we would like to emphasize that we do not aim to compete with the PnP and RED frameworks, but rather we aim to show the benefits of incorporating potential-based denoisers into these frameworks. Notice that our proposed method can be seen as a simple variant of PnP or RED. Nonetheless, following the comments, we will include comparison to versions of DnCNN for which PnP-PGD did not diverge in our experiments (see result summary below).
>
> 2) We did not include a comparison to FISTA-TV, since we compare the performance of our potential-driven recovery scheme with PnP and RED methods that have shown to be superior to standard convex methods, such as FISTA-TV, in many applications. Furthermore, a major contribution of our paper is to show that the potential-driven scheme enjoys theoretical convergence guarantees similar to those of FISTA-TV and other convex optimization methods.
>
> 3) We did observe that PSNR values monotonically improve throughout the iterations until we reach convergence. However, we did not include these results due to the lack of space in the original paper. We accept the reviewer's suggestion and we will include these PSNR plots either in the main body of the supplementary of the revised manuscript.
>
> 4) As our basic deep learning reference we use DnCNN, which is one of the most common and popular denoisers. We chose DnCNN, rather than UNet, since it better handles inputs with arbitrary size. Moreover, notice that the proposed potential network is similar in structure to DnCNN excluding  the fact that we add a few additional operations, such as global average pooling, to ensure that our output is a scalar.
>
> 5) Please note that we include convergence plots showing the difference between two successive intermediate solutions and this difference is equal to the gradient of the objective function. However, we accept the reviewer’s request and we will add convergence plots in terms of the objective function to the revised manuscript.
>
> Image Quality:
> The main benefits of using potential-driven denoisers within PnP and RED frameworks are providing convergence guarantees and removing the sensitivity of the inverse solutions to the particular choice of the step-size. While we observed that PSNR values improve throughout the iterations, we cannot make any claims regarding the final quality. In our experiments, we aim to show that our potential-driven denoisers leads to results that are at least as good as PnP and RED methods with the additional advantage of provable stability.
>
> $\bf Result Analysis$ (PSNR/SSIM)
>
> $\bf Task: Gaussian Deblurring$
>
> $\underline{\text{DnCNN with bias}}$
>
> Set11-              RED:28.18/0.77, PnP:13.41/0.41
>
> BSD500-          RED:27.56/0.72,  PnP:14.77/0.51
>
> $\underline{\text{DnCNN with bias and batch-norm}}$
>
> Set11-             RED:28.41/0.79,        PnP:26.01/0.83
>
> BSD500-        RED:27.81/0.74,        PnP:27.06/0.83
>
> $\underline{\text{DnCNN with batch-norm}}$
>
> Set11-              RED:27.97/0.76,       PnP:23.9/0.79
>
> BSD500-          RED:27.48/0.72,       PnP:26.3/0.81
>
> $\bf Task: Super-Resolution$
>
> $\underline{\text{DnCNN with bias}}$
>
> Set11-            RED:26.09/0.76,       PnP:11.29/0.29
>
> BSD500-       RED:26.17/0.72,        PnP:11.81/0.35
>
> $\underline{\text{DnCNN with bias and batch-norm}}$
>
> Set11-            RED:26.42/0.78,       PnP:22.76/0.73
>
> BSD500-         RED:26.35/0.74,       PnP:23.98/0.72
>
> $\underline{\text{DnCNN with batch-norm}}$
>
> Set11-              RED:26.15/0.76,      PnP:19.6/0.63
>
> BSD500-          RED:26.21/0.73,      PnP:21.73/0.66

---

> > ### Comment · Reviewer_Lp9c · 2021-08-26
> > **Response**
> >
> > Thank you for taking the time to respond to my comments.
> >
> > As I said in my review, I appreciate the concept/idea of designing priors for PnP/RED based on parameterizing a potential function. However, it is unclear why would this additional constraint improve the performance relative PnP/RED. Unfortunately, this paper does not address this important question. Is the improvement due to convergence issues in PnP/RED? Is it due to some form of inherent superiority of the potential-based priors? An issue that further complicates the interpretation of the results is the large performance gap between PnP and RED. What is the reason for big advantage of RED for deblurring and super-resolution?
> >
> > In short, simply presenting an elegant idea is not sufficient for a good paper. I would expect a little bit more technical depth at least via numerical simulations. It would be of value to properly understand the benefits/limitations of training gradient-based deep denoisers relative compared to traditional PnP/RED.

---

> > > ### Author Response · Authors · 2021-09-02
> > > **Author Response to Reviewer Lp9c**
> > >
> > > We appreciate the reviewer’s constructive comments.
> > >
> > > “why would this additional constraint improve the performance relative PnP/RED”:
> > > As stated in the paper, our work is motivated by proximal operators that are basic forms of denoisers for which both RED and PnP converge, and by fact that they must be gradients of some functions (potentials). However, we can provide an alternative explanation to the improvement in terms of convergence and performance by relating our work to MMSE denoisers -
> > > It has been shown (see [51]) that PnP converges with MMSE denoisers, which can be written using the Tweedie’s formula as $$D(x)=x+\sigma^2\nabla_x\log p(x)$$ where $\sigma$ is the noise variance and $\log p(x)$ is the unknown log probability of natural images.
> > > Deep network denoisers such as DnCNN are often used as an approximation for MMSE denoiser. However, as can be inferred from the Tweedie’s formula, such approximations must have symmetric Jacobians, while most common powerful denoisers do not satisfy this condition.
> > > On the other hand, our proposed denoisers are constructed as $$D(x)=x-\nabla g(x)$$ which closely resembles the Tweedie’s formula (for $g(x)\approx-\sigma^2\log p(x)$). Hence, the additional constraints on the denoisers’ architecture creates well-structured/modeled denoisers that are better approximations of MMSE denoisers (and of $\log p(x)$), thus leading to better recovery. This along with the fact we have an explicit potential function allows us to obtain both stability (convergence) and high-recovery performance. Following the reviewer’s comments, we will add the above discussion to the revised manuscript.
> > >
> > > “What is the reason for big advantage of RED for deblurring and super-resolution”:
> > > When performing PnP-PGD or RED-SD with DnCNN one cannot guarantee convergence in general, hence, the iterative processes may or may not diverge. As seen from Fig. 2, at first PnP-PGD performed well, however, beyond a certain point it diverged, leading to bad recovery. Thus, in this case, the performance gap between RED-SD and PnP-PGD is directly related to convergence.  We believe that RED-SD converges in our case since the power of the denoiser is controlled by the regularization parameter, while in PnP-PGD the denoiser is not constrained throughout the iterations.
> > >
> > > “I would expect a little bit more technical depth at least via numerical simulations”:
> > > We understand the reviewer’s concerns. We will incorporate into the revised manuscript a significantly more extensive analysis of our experiments, including the results produced for this rebuttal as well as a thorough study of the different hyperparameters. For fair comparison and for the completeness of the paper, we will extend our discussion on the convergence of PnP-PGD and add results of PnP-PGD performed with early stopping to avoid divergence.

---

### Decision · Program_Chairs · 2021-09-27

**Decision:**

Accept (Poster)

**Comment:**

This paper is proposing a new and interesting idea: a potential-based neural net denoiser that makes RED convergence analysis clean and simple. This is quite novel and suitable to NeurIPS audience. The reviewers raised serious concerns on the evaluation and how the comparisons to other RED methods were performed. I agree that the evaluation should be improved and the authors did a reasonable job on responding to reviewer input on this. I recommend that the authors revise their manuscript taking all this input into account. The analysis on convex functions is also a good suggestion that the authors should take into consideration.